# The Slingshot Mechanism: An Empirical Study of Adaptive Optimizers and the *Grokking Phenomenon*

## Abstract

The *grokking phenomenon* as reported by Power et al. [13] refers to a regime where a long period of overfitting is followed by a seemingly sudden transition to perfect generalization. In this paper, we attempt to reveal the underpinnings of Grokking via a series of empirical studies. Specifically, we uncover an optimization anomaly plaguing adaptive optimizers at extremely late stages of training, referred to as the *Slingshot Mechanism*. A prominent artifact of the Slingshot Mechanism can be measured by the cyclic phase transitions between stable and unstable training regimes, and can be easily monitored by the cyclic behavior of the norm of the last layers weights. We empirically observe that without explicit regularization, Grokking as reported in [13] almost exclusively happens at the onset of *Slingshots*, and is absent without it. While common and easily reproduced in more general settings, the Slingshot Mechanism does not follow from any known optimization theories that we are aware of, and can be easily overlooked without an in depth examination. Our work points to a surprising and useful inductive bias of adaptive gradient optimizers at late stages of training, calling for a revised theoretical analysis of their origin.

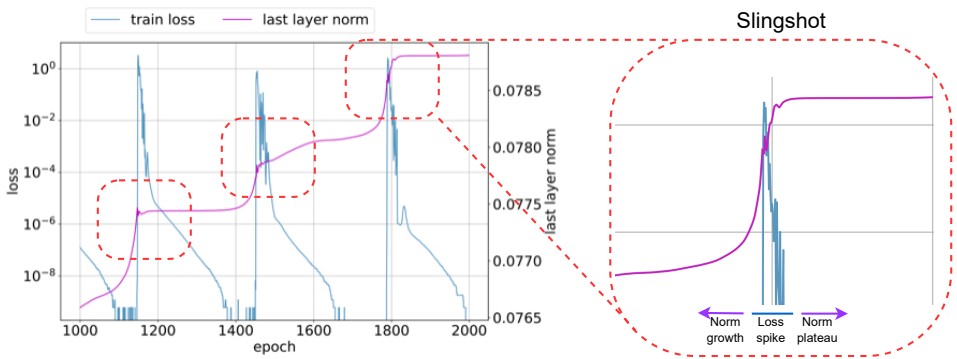

Figure 1: Slingshot Effects are observed with a fully-connected ReLU network (FCN). The FCN is trained with 200 randomly chosen CIFAR-10 samples with Adam. Multiple Slingshot Effects occur in a cyclic fashion as indicated by the dotted red boxes. Each Slingshot Effect is characterized by a period of rapid growth of the last layer weights, an ensuing training loss spike, and a norm plateau.

## 1 Introduction

Recently, the *grokking phenomenon* was proposed by [13], in the context of studying the optimization and generalization aspects in small, algorithmically generated datasets. Specifically, *grokking* refers

Submitted to 36th Conference on Neural Information Processing Systems (NeurIPS 2022). Do not distribute.

to a sudden transition from chance level validation accuracy to perfect generalization, long past the point of perfect training accuracy, i.e., *Terminal Phase of Training* (TPT). This curious behavior contradicts the common belief of early stopping in the overfitting regimes, and calls for further understandings of the generalization behavior of deep neural networks.

In the literature, it has been suggested that in some scenarios, marginal improvements in validation accuracy appears in TPT, which seem to directly support *grokking*. For example, it has been shown in [14] that gradient descent on logistic regression problems converges to the maximum margin solution, a result that has been since extended to cover a wider setting [11, 17]. A key finding in [14] shows that when training on linearly separable data with gradient descent using logistic regression, the classifier's margin slowly improves at a rate of $\mathcal{O}(\frac{1}{\log t})$, while the weight norm of the predictor layer grows at a rate of $\mathcal{O}(t)$, where $t$ is the number of training steps. While specified for gradient descent, Wang et al. [17] showed that similar results also hold for adaptive optimizers (such as Adam and RMSProp). Taking these results into consideration, one could reasonably hypothesise that deep nonlinear networks could benefit from longer training time, even after achieving zero errors on the training set.

In this paper, we provide in depth empirical analyses to the mechanism behind *grokking*. We find that the phenomenology of *grokking* differs from those predicted by [14] in several key aspects. To be concrete, we find that *grokking* occurs during the onset of another intriguing phenomenon directly related to adaptive gradient methods (see Algorithm 1 for a generic description of adaptive gradient methods). In particular, leveraging the basic setup in [13], we make the following observations:

1. During the TPT, training exhibits a cyclic behaviour between stable and unstable regimes. A prominent artifact of this behaviour can be seen in the norm of a model's last layer weights, which exhibits a cyclical behavior with distinct, sharp phase transitions that alternate between rapid growth and plateaus over the course of training.

2. The norm grows rapidly sometime after the model has perfect classification accuracy on training data. A sharp phase transition then occurs when the model missclassifies training samples. This phase change is accompanied by a sudden spike in training loss, and a plateau in the norm growth of the final classification layer.

3. The features (pre-classification layer) show rapid evolution as the weight norm transitions from rapid growth to a growth plateau, and change relatively little at the norm growth phase.

4. Phase transitions between norm growth and norm plateau phases are typically accompanied by a sudden bump in generalization as measured by classification accuracy on a validation set, as observed in a dramatic fashion in [13].

5. It is empirically observed that grokking as reported in [13] almost exclusively happens at the onset of *Slingshots*, and is absent without it.

We denote the observations above as the *Slingshot Effect*, which is defined to be the full cycle starting from the norm growth phase, and ending in the norm plateau phase. And empirically, a single training run typically exhibits multiple Slingshot Effects. Moreover, while *grokking* as described in [13] might be data dependent, we find that the Slingshot Mechanism is pervasive, and can be easily reproduced in multiple scenarios, encompassing a variety of models (Transformers and MLPs) and datasets (both vision, algorithmic and synthetic datasets). Since we only observe Slingshot Effects when training classification models with adaptive optimizers, our work can be seen as empirically characterizing an implicit bias of such optimizers. Finally, while our observations and conclusions hold for most variants of adaptive gradient methods, we focus on Adam in the main paper, and relegate all experiments with additional optimizers to the appendix.

---

**Algorithm 1** Generic Adaptive Gradient Method

---

**Input:** $X_1 \in \mathcal{F}$, step size $\mu$, sequence of functions $\{\phi_t, \psi_t\}_{t=1}^{T}$, $\epsilon \in \mathbb{R}^+$
**Output:** Fitted $\alpha$.
1 **for** $t = 1..., T$ **do**
2     $g_t = \nabla f_t(x_t)$.
3     $m_t = \phi_t(g_1, ..., g_t)$ and $V_t = \psi_t(g_1, ..., g_t)$.
4     $x_{t+1} = x_t - \frac{\mu m_t}{\sqrt{V_t^2} + \epsilon}$

---

## 1.1 Implications of Our Findings

The findings in this paper have both theoretical and practical implications that go beyond characterizing Grokking. A prominent feature of the Slingshot Mechanism is the repeating phase shifts between stable and unstable training regimes, where the unstable phase is characterized by extremely large gradients, and spiking training loss. Furthermore, we find that learning at late stages of training have a cyclic property, where non trivial feature adaptation only takes place at the onset of a phase shift. From a theoretical perspective, this is contradictory to common assumptions made in the literature of convergence of adaptive optimizers, which typically require $L$ smooth cost functions, and bounded stochastic gradients, either in the $L_2$ or $L_\infty$ norm, decreasing step sizes and stable convergence [18, 1, 2]. From the apparent generalization benefits of Slingshot Effects, we cast doubt on the ability of current working theories to explain the Slingshot Mechanism.

Practically, our work presents additional evidence for the growing body of work indicating the importance of the TPT stage of training for optimal performance [6, 13, 12].

In an era where the sheer size of models are quickly becoming out of reach for most practitioners, our work suggest focusing on improved methods to prevent excessive norm growth either implicitly through Slingshot Effects or through other forms of explicit regularization or normalization.

## 2 Related Work

The Slingshot Mechanism we uncover here is reminiscent of the *catapult mechanism* described in Lewkowycz et al. [9]. Lewkowycz et al. [9] show that loss of a model trained via gradient descent with an appropriately large learning rate shows a non-monotonic behavior —the loss initially increases and starts decreasing once the model "catapults" to a region of lower curvature —early in training. However, the catapult phenomenon differs from Slingshot Effects in several key aspects. The *catapult mechanism* is observed with vanilla or stochastic gradient descent unlike the Slingshot Mechanism that is seen with adaptive optimizers including Adam [7] and RMSProp [15]. Furthermore, the *catapult phenomenon* relates to a large initial learning rate, and does not exhibit a repeating cyclic behavior. More intriguingly, Slingshot Effects only emerge late in training, typically long after the model reaches perfect accuracy on the training data.

Cohen et al. [3] describe a "progressive sharpening" phenomenon in which the maximum eigenvalue of the loss Hessian increases and reaches a value that is at equal to or slightly larger than $2/\eta$ where $\eta$ is the learning rate. This "progressive sharpening" phenomenon leads to model to enter a regime Cohen et al. [3] call *Edge of Stability* where-in the model shows non-monotonic training loss behavior over short time spans. *Edge of Stability* is similar to the Slingshot Mechanism in that it is shown to occur later on in training. However, *Edge of Stability* is shown for full-batch gradient descent while we observe Slingshot Mechanism with adaptive optimizers, primarily Adam [7] or AdamW [10].

As noted above, the Slingshot Mechanism emerges late in training, typically longer after the model reaches perfect accuracy and has low loss on training data. The benefits of continuing to training a model in this regime has been theoretically studied in several works including [14, 11]. Soudry et al. [14] show that training a linear model on separable data with gradient using the logistic loss function leads to a max-margin solution. Furthermore Soudry et al. [14] prove that the loss decreases at a rate of $O(\frac{1}{t})$ while the margin increases much slower $O(\frac{1}{\log t})$, where $t$ is the number of training steps. Soudry et al. [14] also note that the weight norm of the predictor layer increases at a logarithmic rate, i.e., $O(\log(t))$. Lyu and Li [11] generalize the above results to homogeneous neural networks trained with exponential-type loss function and show that loss decreases at a rate of $O(1/t(\log(t))^{2-2/L})$. This is, where $L$ is defined as the order of the homogenous neural network. Although these results indeed prove the benefits of training models, their analyses are limited to gradient descent. Moreover, the analyses developed by Soudry et al [14] do not predict any phenomenon that resembles the Slingshot Mechanism. Wang et al. [17] show that homogenous neural networks trained with RMSProp [15] or Adam without momentum [17] do converge in direction to the max-margin solution. However, none of these papers can explain the Slingshot Mechanism and specifically the cyclical behavior of the norm of the last layer weights.

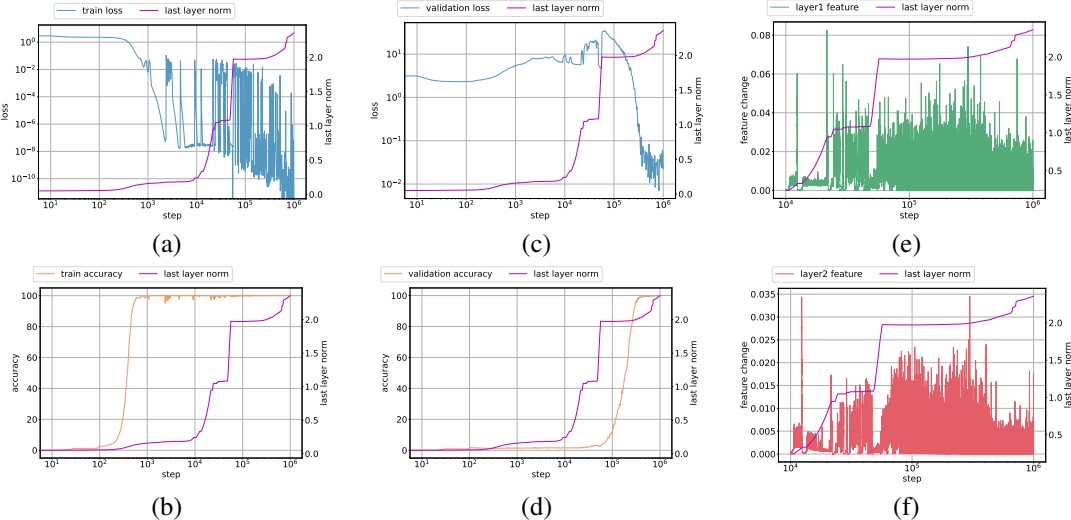

Figure 2: Division dataset: Last layer weight norm growth versus a) loss on training data b) accuracy on training data (c) loss on validation data d) accuracy on validation data e) normalized relative change in features of first Transformer layer (f) normalized relative change in features of second Transformer layer. Note that the feature change plots are shown starting at 10K step to emphasize the feature change behavior during norm growth and plateau phases, revealing that the features stop changing during the norm growth phase and resume changing during the plateaus.

## 3   The Slingshot Mechanism

### 3.1   Experimental Setup

We use the training setup studied by Power et al. [13] in the main paper as a working example to illustrate the Slingshot Mechanism. In this setup, we train decoder-only Transformers [16] on a modular division dataset [13] of the form $a \div b = c$, where $a$, $b$ and $c$ are discrete symbols and $\div$ refers to division modulo $p$ for some prime number $p$, split into training and validation sets. The task consists of calculating $c$ given $a$ and $b$. The algorithmic operations and details of the datasets considered in our experiments are described in Appendix B. The Transformer consists of 2 layers, of width 128 and 4 attention heads with approximately 450K trainable parameters and is optimized by Adam [7, 10]. For these experiments we set learning rate to 0.001, weight decay to 0, $\beta_1 = 0.9$, $\beta_2 = 0.98$, $\epsilon = 10^{-8}$, linear learning rate warmup for the first 10 steps and minibatch size to 512 which are in line with the hyperparameters considered in [13].

Figure 2 shows the metrics of interest that we record on training and validation samples for modular division dataset. Specifically, we measure 1) *train loss*; 2) *train accuracy*; 3) *validation loss*; 4) *validation accuracy*; 5) *last layer norm*: denoting the norm of the classification layer's weights and 6) *feature change*: the relative change of features of the l-th layer ($h^l$) after the t-th gradient update step $\frac{\|h^l_{t+1} - h^l_t\|}{\|h^l_t\|}$. We observe from Figure 2b that the model is able to reach high training accuracy around step 300 while validation accuracy starts improving after $10^5$ steps as seen in Figure 2d. Power et al. [13] originally showed this phenomenon and refer to it as grokking. We observe that while the validation accuracy does not exhibit any change until much later in training, the validation loss shown in Figure 2c exhibits a double descent behavior with an initial decrease, then a growth before rapidly decreasing to zero.

Seemingly, some of these observations can be explained by the arguments in [14] and their extensions to adaptive optimizers [17]. Namely, at the point of reaching perfect classification of the training set, the cross-entropy (CE) loss by design pressures the classification layer to grow in norm at relatively fast rate. Simultaneously, the implicit bias of the optimizer coupled with the CE loss, pushes the direction of the classification layer to coincide with that of the maximum margin classifier, albeit at a much slower rate.

These insights motivate us to measure the classifier's last layer norm during training. We observe in Figure 2a that once classification reaches perfect accuracy on the training set, the classification layer norm exhibits a distinct cyclic behavior, alternating between rapid growth and plateau, with a sharp phase transition between phases. Simultaneously, the training loss retains a low value in periods of rapid norm growth, and then wildly fluctuating in periods of norm plateau. Figure 2e and Figure 2f shows the evolution of the relative change in features output by each layer in the Transformer. We observe that the feature maps are not updated much during the norm growth phase. However, at the phase transition, we observe that the feature maps receive a rapid update, which suggests that the internal representation of the model is updating.

**Is Slingshot a general phenomenon?**  In an attempt to ascertain the generality of Slingshot Effects as an optimization artifact, we run similar experiments with additional architectures, datasets, optimizers, and hyperparameters. We use all algorithmic datasets as proposed in [13], as well as frequently used vision benchmarks such as CIFAR-10 [8], and even synthetic Gaussian dataset. For architectures, we use Transformers, MLPs and deep linear models (see figure 1). We find abundant evidence of Slingshot Effects in all of our experiments with Adam, AdamW and RMSProp. We are unable to observe Slingshot Effects with Adagrad [5] and also with stochastic gradient descent (SGD) or SGD with momentum, pointing to the generality of the mechanism across architectures and datasets. We refer the reader to Appendix A for the full, detailed description of the experiments.

**Why does Slingshot happen?**  We hypothesize that the norm growth continues until the curvature of the loss surface becomes large, effectively "flinging" the weights to a different region in parameter space as small gradient directions get amplified, reminiscent of the mechanics of a slingshot flinging a projectile. We attempt to quantify how far a model is flung by measuring the cosine distance between a checkpoint during optimization and initial parameters. Specifically, we divide the model parameters into representation (pre-classifier) parameters and classifier (last layer) parameters and calculate how far these parameters have moved from initialization. We show that checkpoints collected after a model experiences Slingshot have a larger representation cosine distance. We defer the reader to the appendix for further details.

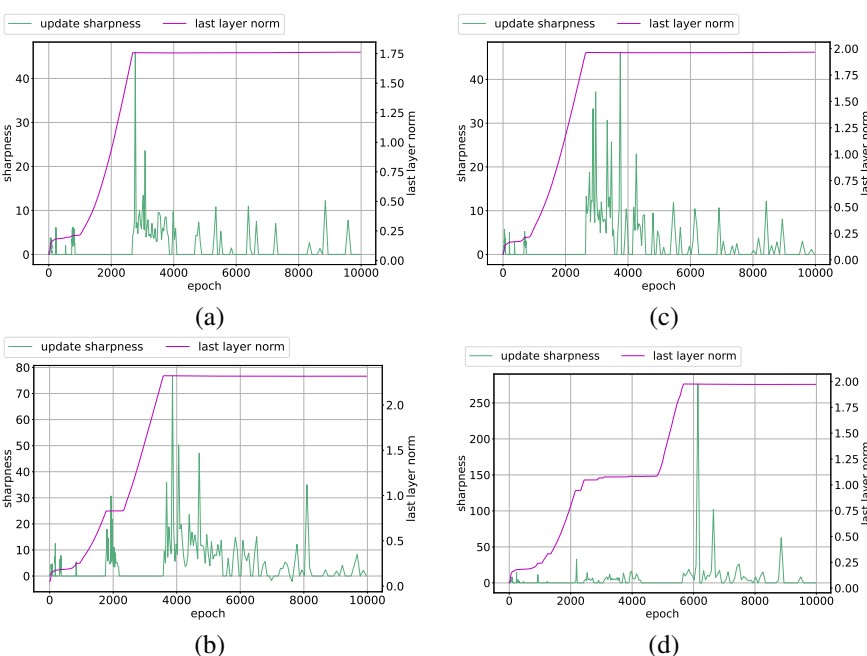

Figure 3: Curvature metric (denoted as "update sharpness") evolution vs norm growth on (a) addition, (b) subtraction, (c) multiplication, and (d) division dataset. Note the spike in the sharpness metric near the phase transitions between norm growth and plateau.

By design, adaptive optimizers adapt the learning rate on a per parameter basis. In toy, convex scenarios, the $\epsilon$ parameter provably determines whether the algorithm will converge stably. To

illustrate this, we take inspiration from [3], and consider a quadratic cost function $\mathcal{L}(A, B, C) = \frac{1}{2}x^\top A x + B^\top x + C$, $A \in \mathcal{R}^{d \times d}$, $x, B \in \mathcal{R}^d$, $C \in \mathcal{R}$, where we assume $A$ is symmetric and positive definite. Note that the global minimum of this cost is given by $x^\star = -A^{-1}B$. The gradient of this cost with respect to $x$ is given by $g = Ax + B$. Consider optimizing the cost with adaptive optimization steps of the simple form $x_{t+1} = x_t - \mu \frac{g}{|g|+\epsilon} = x_t - \mu \frac{Ax_t + B}{|Ax_t + B| + \epsilon}$ where $\mu$ is a learning rate, and the division and absolute operations are taken element wise. Starting from some $x_0$, the error $e_t = x_t - x^\star$ evolves according to:

$$e_{t+1} = \left(I - \mu \text{diag}(\frac{1}{|Ae_t| + \epsilon})A\right)e_t \stackrel{\text{def}}{=} \mathcal{M}_t e_t \tag{1}$$

Note that the condition $\|A\|_s < \frac{2\epsilon}{\mu}$ where $\|\cdot\|_s$ denotes the spectral norm, implies that the mapping $\mathcal{M}_t$ is a contraction for all values of $t$, and hence convergence to the global optimum is guaranteed (This is in contrast to gradient descent, where the requirement is $\|A\|_s < \frac{2}{\mu}$). Note that the choice of $\epsilon$ crucially controls the requirement on the curvature of the cost, represented by the the spectrum of $A$ in this case. In other words, the smaller $\epsilon$, the more restrictive the requirements on the top eigenvalue of $A$. In [3], it was observed that full batch gradient descent increases the spectral norm of the Hessian to its maximum allowed value. We therefore hypothesize that for deep networks, a small value for $\epsilon$ requires convergence to a low curvature local minimum, causing a Slingshot Effect when this does not occur. Moreover, we may reasonably predict that increasing the value of $\epsilon$ would lift the restriction on the curvature, and with it evidence of Slingshot Effects.

Figure 3 shows evidence consistent with the hypothesis that Slingshot Effects occur in the vicinity of high loss curvature, by measuring the local loss surface curvature along the optimization trajectory. Let $\mathcal{H}_t$ denote the local Hessian matrix of the loss, and $u_t$ the parameter update at time $t$ given the optimization algorithm of choice. We use the local curvature along the trajectory of the optimizer, given by $\frac{1}{\|u_t\|^2} u_t^\top \mathcal{H}_t u_t$, as a curvature measure. Across the arithmetic datasets from [13], whenever the last layer weight norm plateaus, the curvature measure momentarily peaks and settles back down.

**Varying $\epsilon$**    We next observe from Figure 2a that the training loss value also spikes up around the time step when the weight norm transitions from growth to plateau. A low training loss value suggests that the gradients (and their moments) used as inputs to the optimizer are small, which in turn can cause the $\epsilon$ hyperparameter value to play a role in calculating updates. Our hypothesis here is that the Slingshot Effect should eventually disappear with a sufficiently large $\epsilon$. To confirm this hypothesis, we run an experiment where we vary $\epsilon$ while retaining the rest of the setup described in the previous section.

Figure 4 shows the results for various values of $\epsilon$ considered in this experiment. We first observe that the number of Slingshot Effect cycles is higher for smaller values of $\epsilon$. Secondly, smaller values of $\epsilon$ cause grokking to appear at an earlier time step when compared to larger values. More intriguingly, models that show signs of grokking also experience Slingshot Effects while models that do not experience Slingshot Effects do not show any signs of grokking. Lastly, the model trained with the largest $\epsilon = 10^{-5}$ shows no sign of generalization even after receiving 500K updates.

## 3.2 Effects on Generalization

In order to understand the relationship between Slingshot Effects and neural networks generalization, we experiment with various models and datasets. We observe that models that exhibit Slingshot tend to generalize better, which suggests the benefit of training models for a long time with Adam [7] and AdamW [10]. More surprisingly, we observe that Slingshots and grokking tend to come in tandem.

**Transformers with algorithmic datasets**    We follow the setting in Power et al. [13] and generate several datasets that represent algorithmic operations and consider several training and validation splits. This dataset creation approach is consistent with the methodology used to demonstrate grokking [13]. The Transformer is trained with AdamW [10] with a learning rate of 0.001, weight decay set to 0, and with learning rate warmup for 500K steps. We consider $\epsilon$ of AdamW as a hyperparameter in this experiment. Figure 5 summarizes the results for this experiment where the x-axis indicates the algorithmic operation followed by the training data split size. As can be seen in Figure 5, Slingshot Effects are seen with lower values of $\epsilon$ and disappear with higher values of $\epsilon$

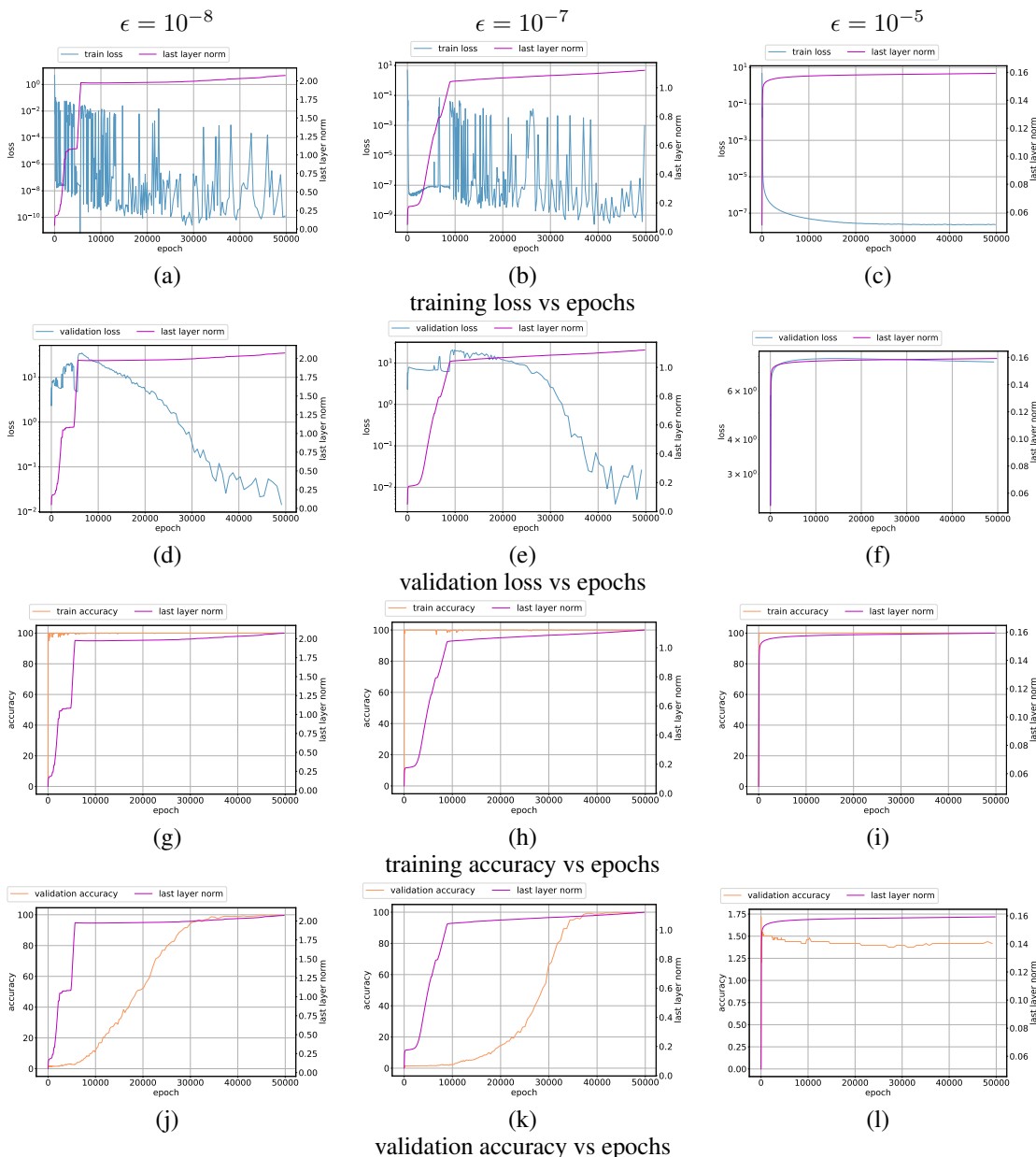

Figure 4: Varying $\epsilon$ in Adam on the Division dataset. Observe that as $\epsilon$ increases, there is no Slingshot Effect or grokking behavior. Figure (a) corresponds to default $\epsilon$ suggested in [7] where the model trained with smallest value undergoes multiple Slingshot cycles.

which confirms the observations made in Section 3 with modular division dataset. In addition, models that exhibit Slingshot Effects and grokking (shown in green) tend to generalize better than models that do not experience Slingshot Effects and grokking (shown in red).

**ViT with CIFAR-10** For further validation of Slingshot Effects and generalization, we train a Vision Transformer (ViT) [4] on CIFAR-10 [8]. The ViT consists of 12 layers, width 384 and 12 attention heads trained on fixed subsets of CIFAR-10 dataset [8]. The ViT model described above is trained with 10K, 20K, 30K, 40K and 50K (full dataset) training samples. We train the models with the following learning rates: 0.0001, 0.00031 and 0.001 and with a linear learning rate warmup for the 1 epoch of optimization. We consider multiple learning rates to study the impact of this hyperparameter on Slingshot taking inspiration from [13] where the authors report observing

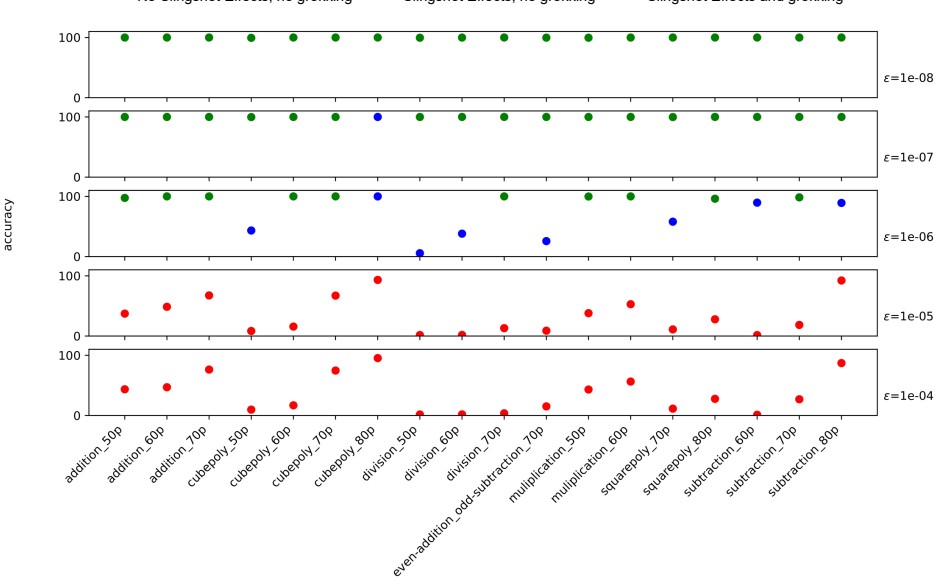

Figure 5: Extended analysis on multiple grokking datasets. Points shown in green represent both Slingshot Effects and grokking, points shown blue indicate Slingshot Effects but not grokking while points in red indicate no Slingshot Effects and no grokking. $\epsilon$ in Adam is varied as shown in text. Observe that as $\epsilon$ increases, there are no Slingshot Effects or grokking behavior.

grokking over a narrow range of learning rates . Figure 6 shows a plot of the highest test accuracy for a set of hyperparameters (learning rate, number of training samples) as a function of the number of training samples from which we make the following observations. The best test accuracy for a given set of hyperparameters is typically achieved after Slingshot phase begins during optimization. The checkpoints that achieve the highest test accuracy are labeled as "post-slingshot" and shown in green in Figure 6. While post-Slingshot checkpoints seem to enjoy higher test accuracy, there are certain combinations of hyperparameters that lead to models that show better test accuracy prior to the start of the first Slingshot phase. We label these points as "pre-slingshot" (shown in blue) in Figure 6. The above observations appear to be consistent with our finding that training long periods of time may lead to better generalization seen with grokking datasets [13].

**Non-Transformer Models**   We conduct experiments with MLPs on synthetic data where the synthetic data is a low dimensional embedding projected to higher dimensions via random projections. This design choice is critical with showing the existence of the Slingshot Effect with synthetically generated data. We find that using low dimensional data does not lead to any Slingshots. With this dataset, we show that generalization occurs late in training with Adam. Specifically, we tune $\epsilon$ in Adam and show that the optimizer is highly sensitive to this hyperparameter. These observations are consistent with the behavior reported above with Transformers and on algorithmic datasets as well as standard vision benchmark such as CIFAR-10. We refer the reader to Appendix **??** for complete description and details of these experiments.

### 3.3   Drawbacks and Limitations

While the Slingshot Mechanism exposes an interesting implicit bias of Adam that often promotes generalization, due to its arresting of the norm growth and ensuing feature learning, it also leads to some training instability and prolonged training time. In the Appendix we show that it is possible to achieve similar levels of generalization with Adam on the modular division dataset [13] using the same Transformer setup as above, while maintaining stable learning, in regimes that do not show a clear Slingshot Effect. First we employ weight decay, which causes the training loss values to converge to a higher value than the unregularized model. In this regime the model does not become unstable, but instead regularization leads to comparable generalization, and much more quickly. However, it is important to tune the regularization strength appropriately. Similarly, we find that it is

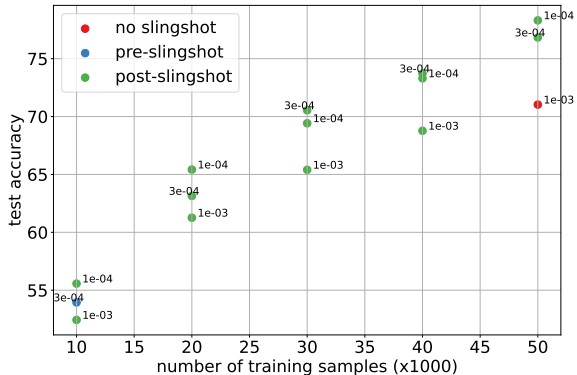

Figure 6: Slingshot Effects on subsets of CIFAR-10 dataset. We train ViTs with multiple learning rates to verify the impact this parameter has on Slingshot. Power et al [13] note that grokking occurs over a narrow range of learning rates. Note that the points marked in: (i) green correspond to test accuracy for an experiment after the Slingshot Effect begins, (ii) blue are for trials where best checkpoint is observed prior to start of a Slingshot Effect and (iii) red are for trials with no Slingshot Effect.

possible to normalize the features and weights using the following scheme to explicitly control norm growth: $w = \frac{w}{\|w\|}, f(x) = \frac{f(x)}{\|f(x)\|}$, where $w$ and $f(x)$ are the weights and inputs to the classification layer respectively, the norm used above is the $L_2$ norm, and $x$ is the input to the neural network. This scheme also results in stable training and similar levels of generalization. In all cases the effects rely on keeping the weight norms from growing uncontrollably, which may be the most important factor for improving generalization. These results suggest that while the Slingshot Mechanism may be an interesting self-correcting scheme for controlling norm growth, there are likely more efficient ways to leverage adaptive optimizers to similar levels of generalization without requiring the instability that is a hallmark of the Slingshot effect.

Finally, we lack a satisfactory theoretical explanation for the Slingshot Mechanism, and hence removed all attempts at a more rigorous mathematical definition, which we feel would only serve as a distraction.

## 4   Conclusion

We have empirically shown that optimizing deep networks with cross entropy loss and adaptive optimizers produces the Slingshot Mechanism, a curious optimization anomaly unlike anything described in the literature. We have provided ample evidence that Slingshot Effects can be observed with different neural architectures and datasets. Furthermore, we find that Grokking [13] almost always occurs in the presence of Slingshot Effects and associated regions of instability in the Terminal Phase of Training (TPT). These results in their pure form absent explicit regularization, reveal an intriguing inductive bias of adaptive gradient optimizers that becomes salient in the TPT, characterized by cyclic stepwise effects on the optimization trajectory. These effects often promote generalization in ways that differ from non-adaptive optimizers like SGD, and warrant further study to be able to harness efficiently. There are open question remaining to be answered, for instance **1)** What's the causal factor of the plateau of weight norm growth? **2)** Are there better ways of promoting generalization without relying on this accidental training instability? Answering these questions w ill allow us to decouple optimization and regularization, and ultimately to control and improve them independently.

## 5   Societal Impact

This is a fundamental work in Deep Learning, it will impact the society via its effects on relevant models and applications.

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
