# The Slingshot Mechanism: An Empirical Study of Adaptive Optimizers and the *Grokking Phenomenon* - Appendix

# Contents

## A  Slingshot Effects across Architectures, Optimizers and Datasets

This section provides further evidence of the prevalence of Slingshot across architectures and optimizers on subsets of CIFAR-10, testing setups beyond the specific setup consider by Power et al. [12]. In these experiments, we focus solely on characterizing the optimization properties of various setups described below. The small sample sizes are used in order to more easily find regimes where different architectures can converge to fit the training data fairly quickly.

We use cross-entropy loss to optimize the models with AdamW [9] in the following experiments. The following experiments are implemented in PyTorch [10].

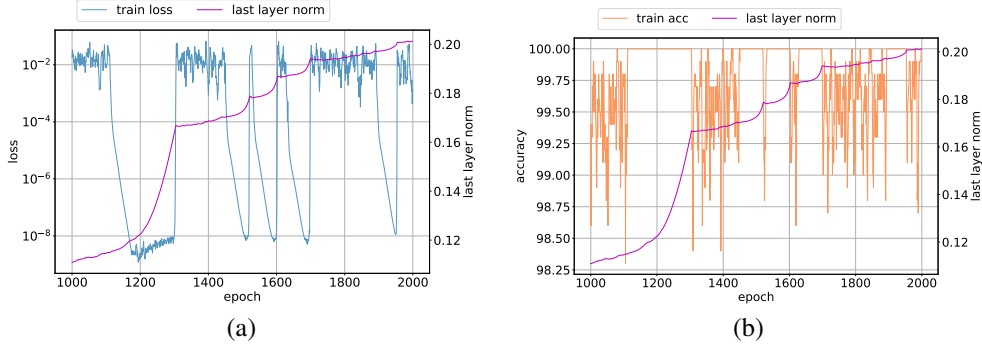

Figure 1: Vision Transformer on 1000 samples from CIFAR-10: Norm growth versus a) loss on training data b) accuracy on training data for a ViT trained on 1000 samples.

### A.1  Vision Transformers on 1000 samples from CIFAR-10

For further validation, we train a Vision Transformer (ViT) [5] with 12 layers that has 10 million parameters on a small sample of the CIFAR-10 dataset [8]. In this setup, we use a learning rate to 0.001, no weight decay, $\beta_1 = 0.9$, $\beta_2 = 0.95$, $\epsilon = 10^{-8}$ and minibatch size of 128. We choose a sample size of 1000 training samples for computational reasons, as we wish to observe multiple cycles of the Slingshot Mechanism extremely late in training. The input images are standardized to be in the range $[0, 1]$. No data augmentation is used in our training pipeline. Due to the extremely small sample size, we focus our attention on the training metrics since no generalization is expected. Figure 1a (respectively Figure 1b) shows a plot of training loss (respectively training accuracy) and last layer norm evolution during the latter stages of training. Multiple Slingshot stages are observed in these plots (5 clear cycles), which can be seen by the sharp transition of the weight norm from high growth to plateau.

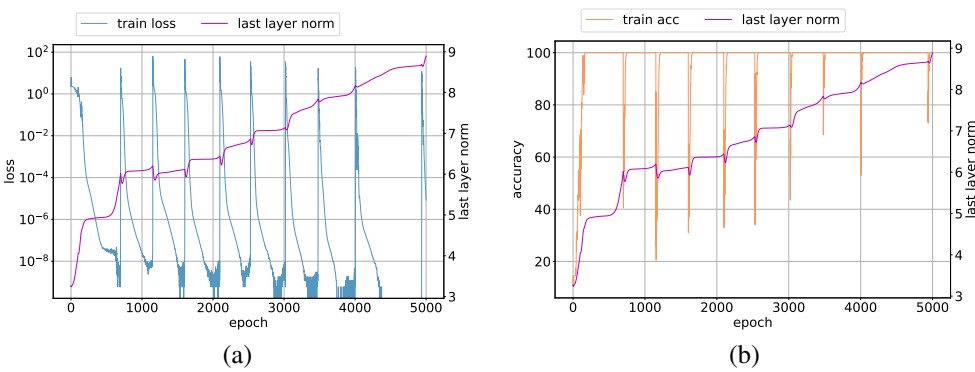

Figure 2: CNN on CIFAR-10 dataset: Norm growth versus a) loss on training data b) accuracy on training data for a VGG11-like model without batch normalization trained on 200 samples.

## A.2 CNN on 200 samples from CIFAR-10

We consider a VGG-like architecture [13] that has been adapted for CIFAR-10 dataset.[1] The model is trained with 200 randomly chosen samples from CIFAR-10 training split and with full-batch AdamW [9]. The hyperparameters used for the optimizer include a learning rate of $0.001$, weight decay$= 0$, $\beta_1 = 0.9$, $\beta_2 = 0.95$, and $\epsilon = 10^{-8}$. As with ViT, no data augmentation is used in these experiments other than standardizing the input to be in the range $[0, 1]$. We observe the presence of multiple Slingshot stages with CNN from Figure 2a and Figure 2b. These experiments suggest that Slingshot effect is not restricted to Transformers architecture alone.

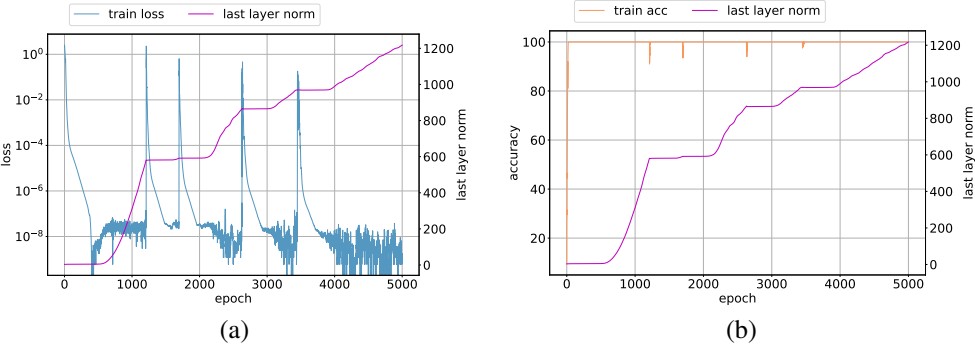

(a)  (b)

Figure 3: CNN on 200 samples from CIFAR-10: Norm growth versus a) loss on training data b) accuracy on training data for a VGG11-like model without batch normalization trained on 200 samples.

**With BatchNorm** We repeat the CNN-based described above but with a VGG-like model that includes batch normalization [6].[2] The training setup is identical to the one described for CNN wihtout batch normalization. We observe the presence of multiple Slingshot stages with CNN from Figure 3a and Figure 3b. The weight norm does not decrease during training as opposed to the weight norm dynamics for CNN wihtout batch normalization seen in Figure 2. These experiments suggest that Slingshot Effects can be seen with standard neural network training components including batch normalization.

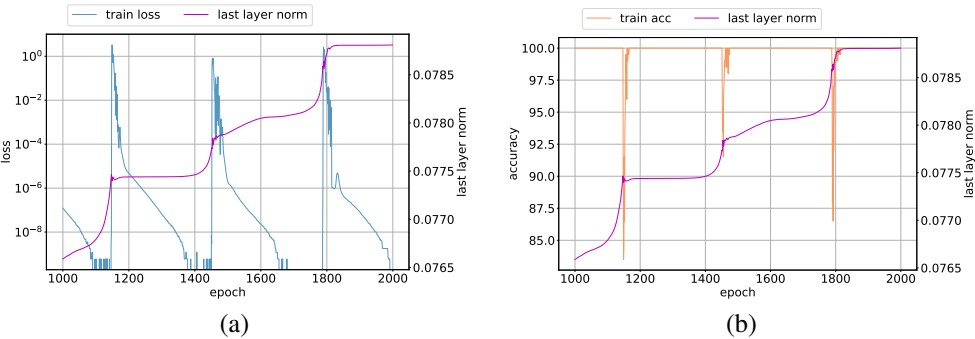

(a)  (b)

Figure 4: MLP on 200 samples from CIFAR-10: Norm growth versus a) loss on training data b) accuracy on training data for a model trained on 200 samples.

## A.3 MLPs on 200 samples from CIFAR-10

The next architecture we consider is a deep (6 layers) fully connected network trained on a small sample of 200 samples belonging to the CIFAR-10 dataset [8] with full-batch AdamW [9] optimizer. The optimizer's hyperparameters are set as following: learning rate $= 0.001$, weight decay $= 0$,

---

[1]We use the VGG11 architecture without batch normalization [6] from https://github.com/kuangliu/pytorch-cifar in this experiment.

[2]We use the VGG11 architecture with batch normalization [6] from https://github.com/kuangliu/pytorch-cifar in this experiment.

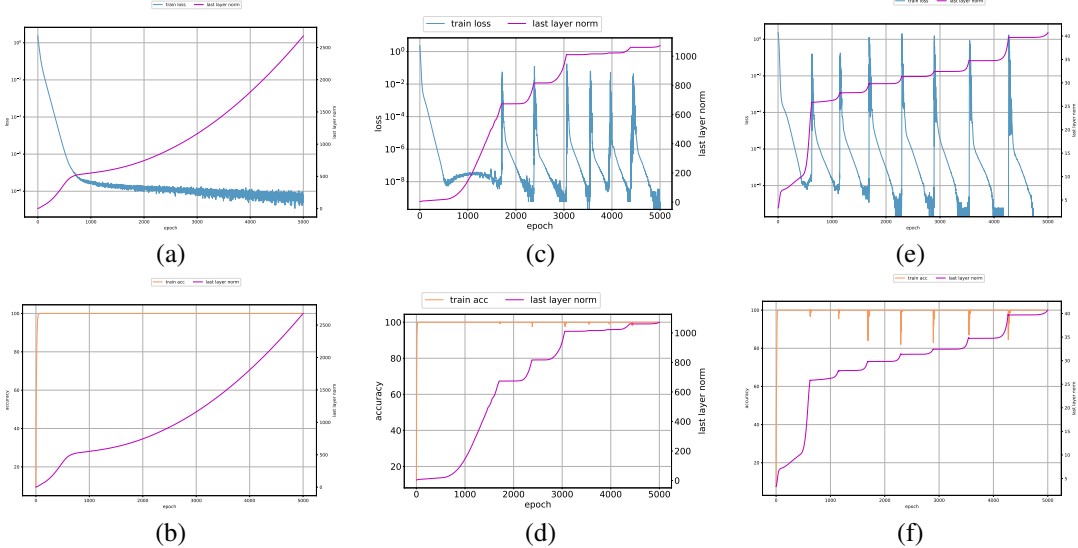

Figure 5: Training shallow models on 200 samples from CIFAR-10: (a) Training loss for 1 layer (linear) model (b) Training accuracy for 1 layer (linear) model (c) Training loss for 2 layer MLP (d) Training accuracy for RMSProp (e) Training loss for 3 layer MLP and (f) Training accuracy for 3 layer MLP . All models are trained with full-batch Adam with learning rate 0.001 on 200 CIFAR-10 samples.

$\beta_1 = 0.9$, $\beta_2 = 0.95$, and $\epsilon = 10^{-8}$. As with the ViT setup above we do no use data augmentation for training this model. Figure 4a (respectively Figure 4b) shows a plot of training loss (respectively training accuracy) and last layer norm evolution during the latter stages of training. Multiple Slingshot stages are observed in this setup as well. These experiments further suggest that the Slingshot mechanism is prevalent in simple models as well.

## A.4 Shallow models

We consider the behavior of shallow models including linear, 2- and 3-layer MLPs with Adam optimizer. As with the previous setup, we train these models on a small sample of 200 samples belonging to the CIFAR-10 dataset [8] with full-batch Adam [7] optimizer. The optimizer's hyperparameters are set as following: learning rate = 0.001, weight decay = 0, $\beta_1 = 0.9$, $\beta_2 = 0.95$, and $\epsilon = 10^{-8}$. No data augmentation is used in these experiments as well. Figure 5a, Figure 5c, Figure 5e show the training loss and last layer norm evolution during training for the linear, 2-layer and 3-layer models respectively while Figure 5b, Figure 5d, Figure 5f show the training accuracy and last layer norm evolution. Slingshot Effects are observed in 2-layer and 3-layer MLPs whereas no Slingshot Effects are seen with the linear model. These experiments suggest that depth appaears to be a necessary condition to observe Slingshots.

## A.5 Deep linear models

We train a 6 layer linear model with 200 samples belonging to CIFAR-10 [8] with full-batch AdamW [9]. The optimizer's hyperparameters are set as following: learning rate = 0.001, weight decay = 0, $\beta_1 = 0.9$, $\beta_2 = 0.95$, and $\epsilon = 10^{-8}$. Figure 6a and Figure 6b show the training loss and accuracy behavior observed during optimization. Multiple Slingshot stages are observed with this architecture as well.

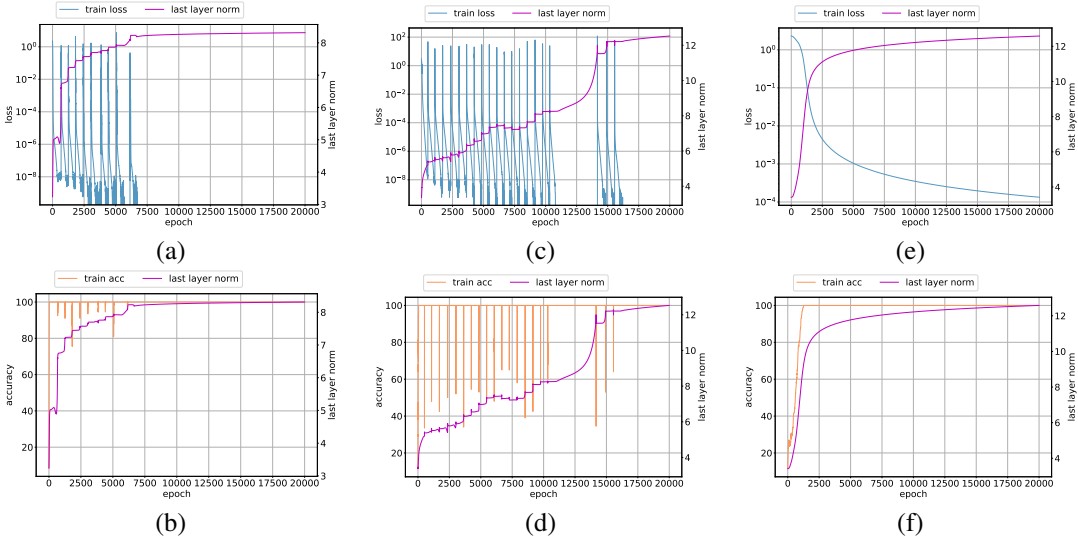

Figure 6: Optimizer choice on deep linear models on 200 samples from CIFAR-10: (a) Training loss for AdamW (b) Training accuracy for AdamW (c) Training loss for RMSProp (d) Training accuracy for RMSProp (e) Training loss for Gradient Descent and (f) Training accuracy for Gradient Descent. All optimizers train a 6-layer linear model full-batch on 200 CIFAR-10 samples.

## A.6 Learning Subset Parities

In this section we use the $k$-sparse parities of $n$ bits task as a test bed. Theoretically, this family of tasks is notoriously challenging since it poses strict computational lower bounds on learning (see for more details). For the $(k, n)$ subset parity task, each input is a random $n$ dimensional vector such that each component is randomly sampled from $\sim \text{Unif}\{-1, 1\}$. The label is then given by a parity function over a predefined sparse set of $k \ll n$ bits. For the following experiments, we use $k = 3, n = 50$. For the model, we use a 3 layer MLP with $relu$ activations, and the cross entropy loss. We use a dataset of 1000 samples, and a test set of 8000 samples. We train each network with Adam using a batch size of 32, a learning rate of $\eta = \{0.004, 0.003, 0.002\}$ and $\epsilon \in \{10^{-8}, 10^{-7}, 10^{-6}\}$. Our results are summarized in figures 7, 8 and 9. For $\epsilon = 10^{-8}$, multiple Slingshots appear past the perfect fitting of the training set, with a bump in generalization post most Slingshots. For larger values of $\epsilon$, no Slingshots are observed, while generalization remains poor.

### A.6.1 Effective Step Size and Curvature Dynamics

A classical results pertaining to optimizing smooth functions with gradient descent states that a sufficient condition for convergence requires that the learning rate does not exceed $\frac{2}{L}$, where $L$ is the Lipschitz constant of the gradient. Due to the sufficiency of the condition, we expect it to be violated at the phase transitions of the slingshots, when the training loss spikes. We quantify the effective step size of a parameter as $\frac{\eta}{\sqrt{V_t^2 + \epsilon}}$ where the terms are defined in Algorithm 1. To approximate $L$ in a local region, we use the maximum eigenvalue of the loss Hessian in this analysis as is done by a series of recent works including Cohen et al. [4], Ahn et al. [1] and Arora et al. [2]. We use the same setup described for training parity dataset to conduct this empirical analysis. The hyperparameters used for the optimizer include $\eta = 0.004$, $\epsilon = 10^{-8}$ and $\beta_1 = 0.9$ and $\beta_2 = 0.999$. Figure 10a shows the dynamics of the training and validation loss while Figures 10b, Figure 10c and Figure 10d shows the evolution of the effective step size as well as the maximum allowable step size for a few parameters chosen randomly from the three layers in the neural network. We observe from these plots that the effective step size is smaller than the maximum allowed step size in the vicinity of SlingShot Effects. however, at the phase transitions we clearly see that the effective step size is larger than the maximum allowed, causing the loss to spike. After a few Slingshot cycles, we observe that the maximum allowed step size increase dramatically, and no additional Slingshots follow.

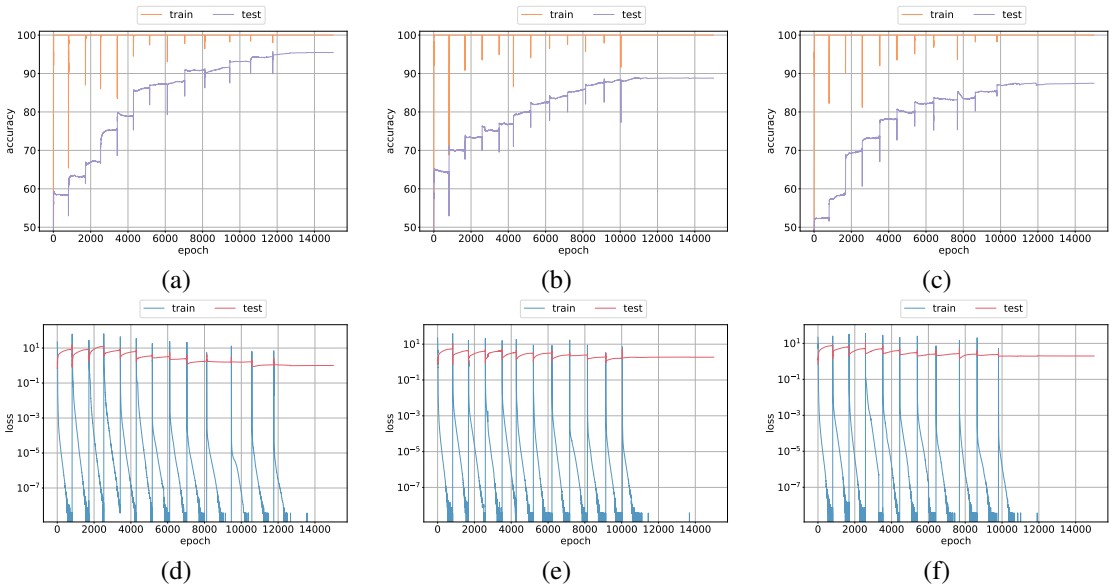

Figure 7: Learning a $(3, 50)$ subset parity with Adam with $\epsilon = 10^{-8}$ and a learning rate of (a),(d) $\eta = 0.004$, (b),(e) $\eta = 0.003$ and (c),(f) $\eta = 0.002$. Multiple Slingshots are visible, resulting in improved generalization.

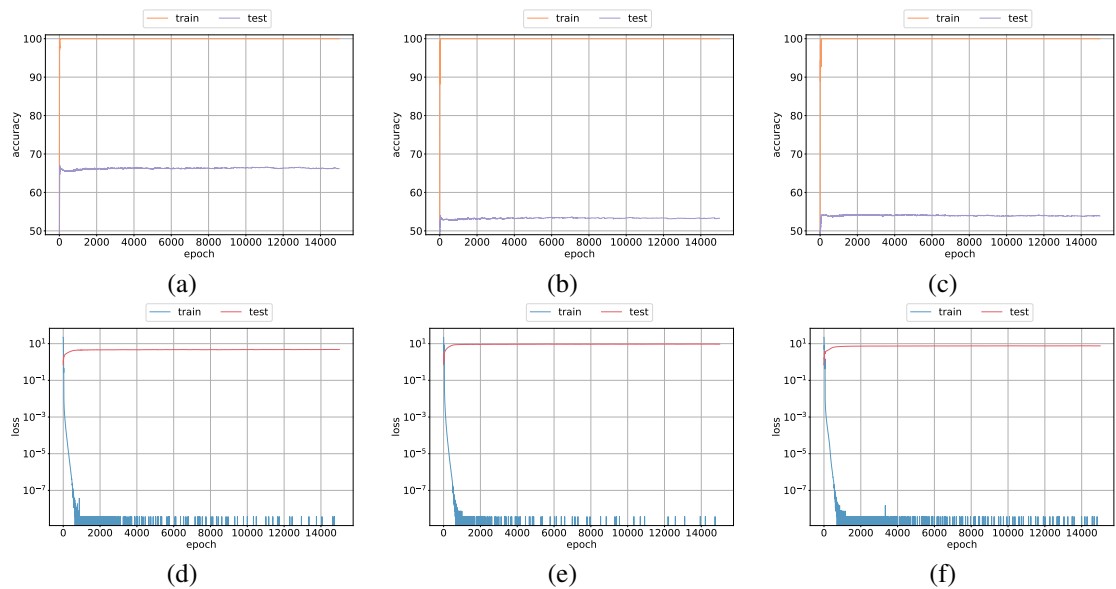

Figure 8: Learning a $(3, 50)$ subset parity with Adam with $\epsilon = 10^{-7}$ and a learning rate of (a),(d) $\eta = 0.004$, (b),(e) $\eta = 0.003$ and (c),(f) $\eta = 0.002$. No Slingshots are visible.

## A.7 Different Optimizers

In this set of experiments, we study the training loss behavior of deep linear models optimized full-batch with AdamW [9], RMSProp [14] and full-batch gradient descent (GD). The six layer model is trained with 200 samples. The hyperparameters used for optimizing the model with various optimizers are described in Table 1. Figure 6 shows the training loss and accuracy behavior of the three optimizers considered in this experiment. We observe Slingshot behavior with AdamW and RMSProp from Figure 6 while Slingshot behavior is absent with standard gradient descent. This observation suggests that the normalization used in adaptive optimizers to calculate the update from gradients may lead to Slingshot behavior.

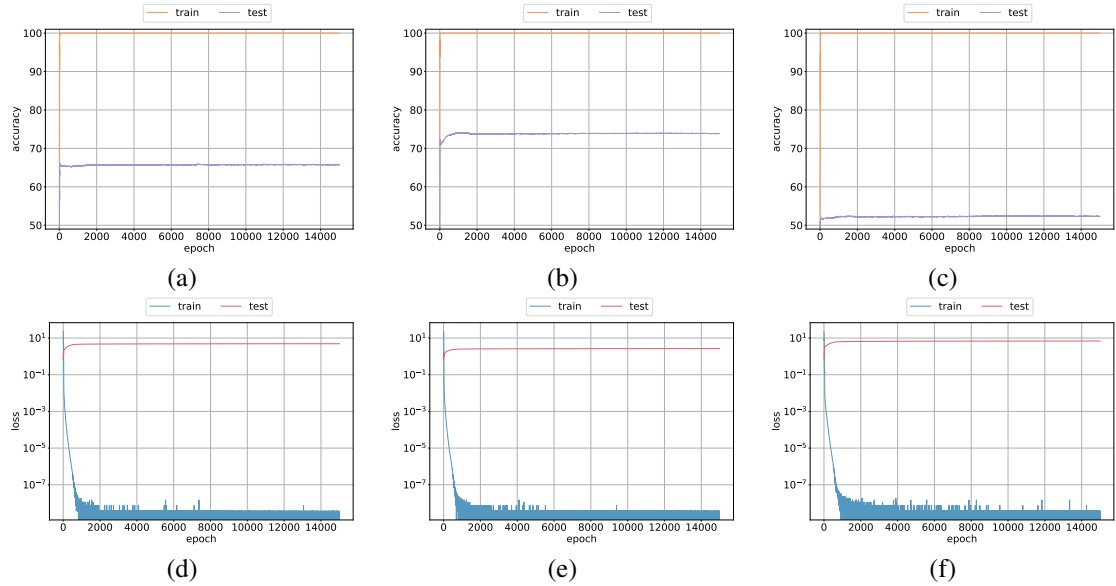

Figure 9: Learning a $(3, 50)$ subset parity with Adam with $\epsilon = 10^{-6}$ and a learning rate of (a),(d) $\eta = 0.004$, (b),(e) $\eta = 0.003$ and (c),(f) $\eta = 0.002$. No Slingshots are visible.

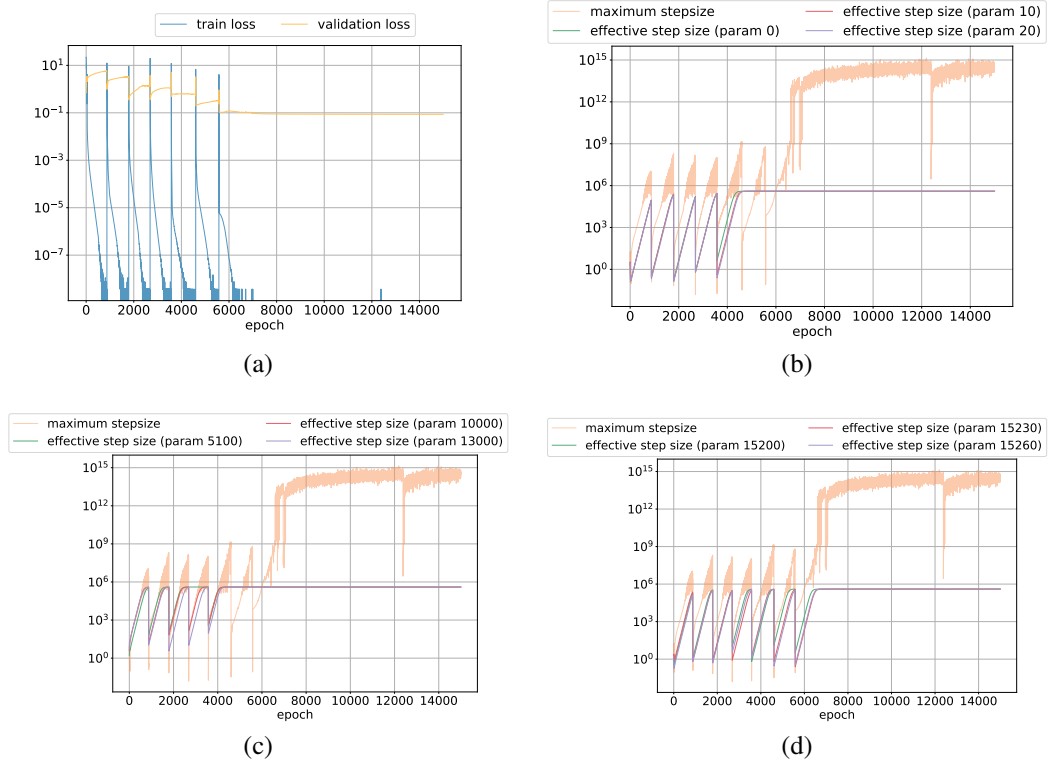

Figure 10: Empirical analysis of the relationship between Slingshot Effects and loss surface sharpness. Above plots include (a) training and validation loss; evolution of effective step size and curvature of parameters from (b) first layer, (c) second layer and (d) classification layer in a 3-layer MLP trained with Adam. At the phase transitions, effective step size is larger than $\frac{2}{L}$, initiating the slingshots. After a few cycles, the Lipschitz constant of the gradients decreases substantially, and the Slingshots cease.

Table 1: Optimizers hyperparameters. Learning rate is set to 0.001 and weight decay to 0 for all optimizers

| Optimizer | Other hyperparameters |
|-----------|----------------------|
| Adam | $\beta_1 = 0.9, \beta_2 = 0.95$ |
| RMSProp | $\alpha = 0.95$, momentum=0.0 |
| GD | momentum=0.9 |

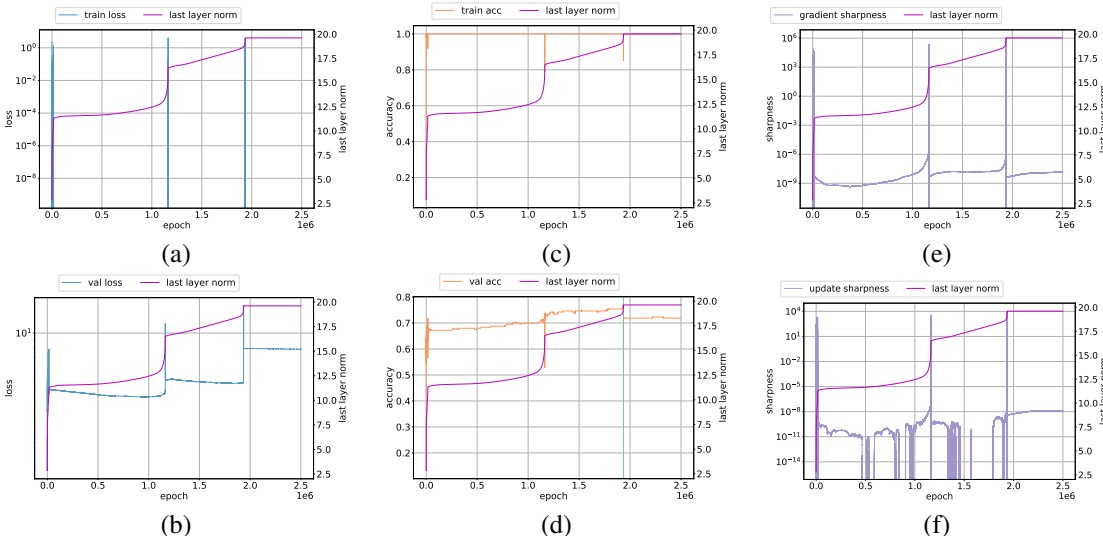

Figure 11: Slingshot generalization on synthetic dataset: Norm growth versus a) loss on training data b) accuracy on training data (c) loss on validation data d) accuracy on validation data. Note that the vertical line in green shows location of maximum test accuracy. Adam hyperparameters are $\beta_1 = 0.9, \beta_1 = 0.95, \epsilon = 10^{-8}$

## A.8 Slingshot with MLP and Synthetic Dataset

In this section, we provide empirical evidence that Slingshot Effects are observed with a synthetic dataset in a fully-connected architecture. The small dimensional dataset, like the Grokking dataset of Power et al. [12], allows us to easily measure of sharpness, given by $\frac{1}{\|u_t\|^2} u_t^\top \mathcal{H}_t u_t$ where $u_t$ is the optimizer's update vector and $\mathcal{H}_t$ is the Hessian at step $t$, to examine the interplay between Slingshot Effects and generalization.

**Vision Transformers and Full CIFAR-10**  In Appendix A, we have empirically shown that the existence of the Slingshot phenomenon on a small subset of CIFAR-10 dataset [8] with Vision Transformers (ViTs). We now study the impact that Slingshot has on the generalization ability of ViTs by training a model on all 50000 samples in CIFAR-10 training dataset. The ViT used here is a larger model than the one considered in A to account for larger dataset size. The ViT model consists of 12 layers, width 384 and 12 attention heads and is optimized by AdamW [9]. For this experiment, we set the learning rate to 0.0001, weight decay to 0, $\beta_1 = 0.9$, $\beta_1 = 0.95$ and $\epsilon = 10^{-8}$, minibatch size of 512 and linear learning rate warmup for 1 epoch of optimization. Figure 12 shows the results of experiment with full CIFAR-10 dataset. Multiple Slingshots can be observed in these plots similar to the plots described in Appendix A. We observe from Figure 12d that the test accuracy peaks in epochs following a Slingshot with the maximum recorded test accuracy occurring very late in optimization. This observation suggests that the Slingshot can have a favorable effect on generalization consistent with the behavior observed in the main paper with division dataset.

### A.8.1 Abalation Study

In this section, we train a toy model on a synthetically generated dataset with the aim of analysing the effect of different hyper parameters on the Slingshot Mechanism. We construct a 128-dimensional dataset with Scikit-learn [11] that has 3 informative dimensions that represents a 8-class classification problem. The class centers are the edges of a 3-dimensional hypercube around which clusters are

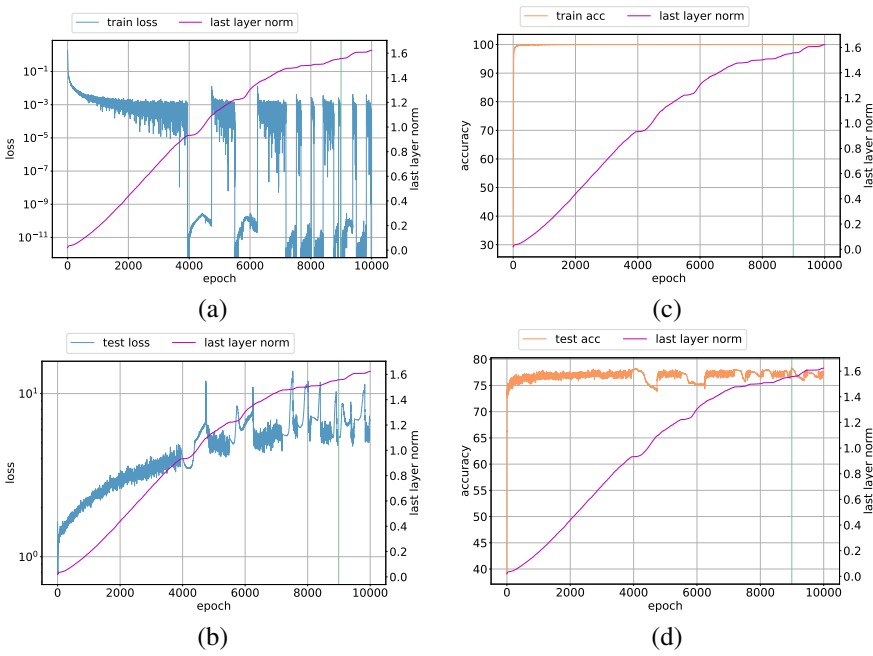

Figure 12: Slingshot generalization on full CIFAR-10 dataset: Norm growth versus a) loss on training data b) accuracy on training data (c) loss on test data d) accuracy on test data.

data are sampled from a standard normal distribution. The other 125-dimensions are also filled at random to create a high-dimensional dataset used in our experiments. We generate 256 training and validation samples for this dataset and use a minibatch size of 128 in all the experiments described in the following.

**Architecture and Optimizer**   Figure 11 shows the training and validation metrics when we optimize a 4-layer fully-connected network (FCN) with Adam using a learning rate of 0.001, $\beta_1 = 0.9$, $\beta_1 = 0.95$, no weight decay and $\epsilon = 10^{-8}$. Note that we use this value of $\epsilon$ in our first experiment as this is the default value proposed in Kingma and Ba [7]. These experiments are implemented in JAX [3].

**Tuning $\epsilon$**   In the next set of experiments with synthetic data, we tune $\epsilon$ value for Adam to understand its impact on test accuracy. Figure 13 shows a plot of the maximum validation accuracy achieved by models trained with Adam as a function of time (epoch). We observe that Adam reaches its best test accuracy late in optimization with $\epsilon = 10^{-5}$ yielding the highest validation accuracy. Furthermore, the best accuracy is achieved with a model that experiences Slingshot during optimization. This observation is consistent with our findings for ViT training with CIFAR-10 dataset described in the main paper and Appendix A.8.

**Influence of $\beta_1$ and $\beta_2$**   In these experiments, we aim to study the impact of Adam/AdamW optimizer's $\beta_1$ and $\beta_2$ hyperparameters on Slingshot. We use the synthetic data described above and set the learning rate of 0.001 and $\epsilon = 10^{-8}$ for this analysis. Figure 14 and Figure 15 shows the results of this study. We observe from Figure 14 that the Slingshot Mechanism is fairly robust to the values of $\beta_1$ and $\beta_2$. Figure 14a-Figure 14c show that Slingshot is even observed with $\beta_1$ and $\beta_2$ set to 0 which effectively disables exponential moving averaging of gradient moments in Adam [7]. Figure 14g-Figure 14i provide an example of hyperparameters that fail to induce Slingshot. We observe from Figure 14 that models that experience Slingshot tend to reach their best test accuracy during the later stages of training. Specifically, we observe from Figure 14b, Figure 14e and Figure 14k that the best validation accuracy occurs after 60000 epochs. These examples provide further evidence about an interesting implicit bias of Adam. Figure 15 shows more examples of hyperparameters that do not induce Slingshot Effects. Finally, we observe from Figure 15 that hyperparameters that provide higher validation accuracy are from models that experience Slingshot Effects.

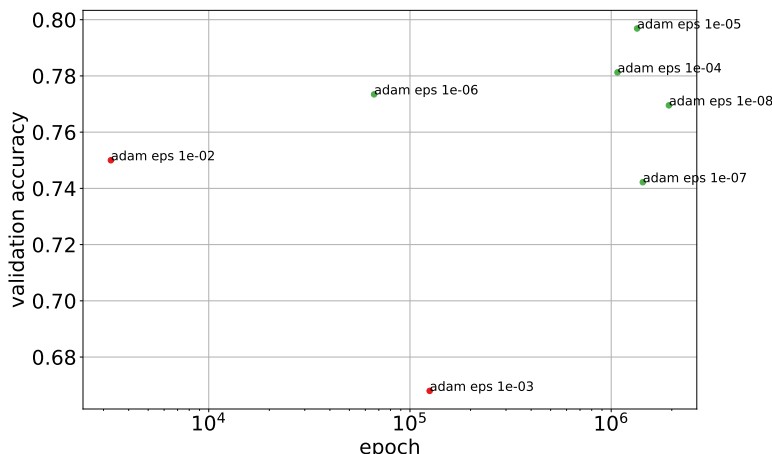

Figure 13: Slingshot on synthehtic dataset. Note that the points marked in: (i) green correspond to Adam-trained models that undergo Slingshot, (ii) red correspond to Adam-trained models that do not experience Slingshot; Adam's hyperparameters are given by $\beta_1 = 0.9$, $\beta_2 = 0.95$, no weight decay and $\epsilon$ shown in parentheses.

## B   Slingshot and Grokking

We use the empirical setup described by Power et al. [12] to describe the Slingshot Mechanism. The following section describes relevant details including datasets, architecture and optimizer used in our experiments.

**Architecture**   The model used a decoder-only Transformer [15] with causal attention masking. The architecture used in all our experiments consists of 2 decoder layers with each layer of width 128 and 4 attention heads.

**Optimization**   We train the architecture described above with Adam optimizer [7, 9] in most of our experiments unless noted otherwise. The learning rate is set to 0.001 and with linear learning rate warmup for the first 10 steps. We use $\beta_1 = 0.9$, $\beta_2 = 0.98$ for Adam's hyperparameters. The Transformers are optimized with cross-entropy (CE) loss that is calculated on the output tokens for a given binary operation.

**Algorithmic Datasets**   The Transformer is trained on small algorithmic datasets that consists of sequences that represent a mathematical operation. The following operations are used in our experiments:

$c = a + b \pmod{p}$ for $0 \leq a, b < p$

$c = a - b \pmod{p}$ for $0 \leq a, b < p$

$c = a * b \pmod{p}$ for $0 \leq a, b < p$

$c = a \div b \pmod{p}$ for $0 \leq a, b < p$

$c = a^2 + b \pmod{p}$ for $0 \leq a, b < p$

$c = a^3 + b \pmod{p}$ for $0 \leq a, b < p$

$c = a^2 + b^2 \pmod{p}$ for $0 \leq a, b < p$

$c = a^2 + b^2 + ab \pmod{p}$ for $0 \leq a, b < p$

$c = a^2 + b^2 + ab + b \pmod{p}$ for $0 \leq a, b < p$

$c = a^3 + ab \pmod{p}$ for $0 \leq a, b < p$

$c = a^3 + ab^2 + b \pmod{p}$ for $0 \leq a, b < p$

$c = [a \div b \pmod{p}$ if $b$ is odd, otherwise $a - b \pmod{p}]$ for $0 \leq a, b < p$

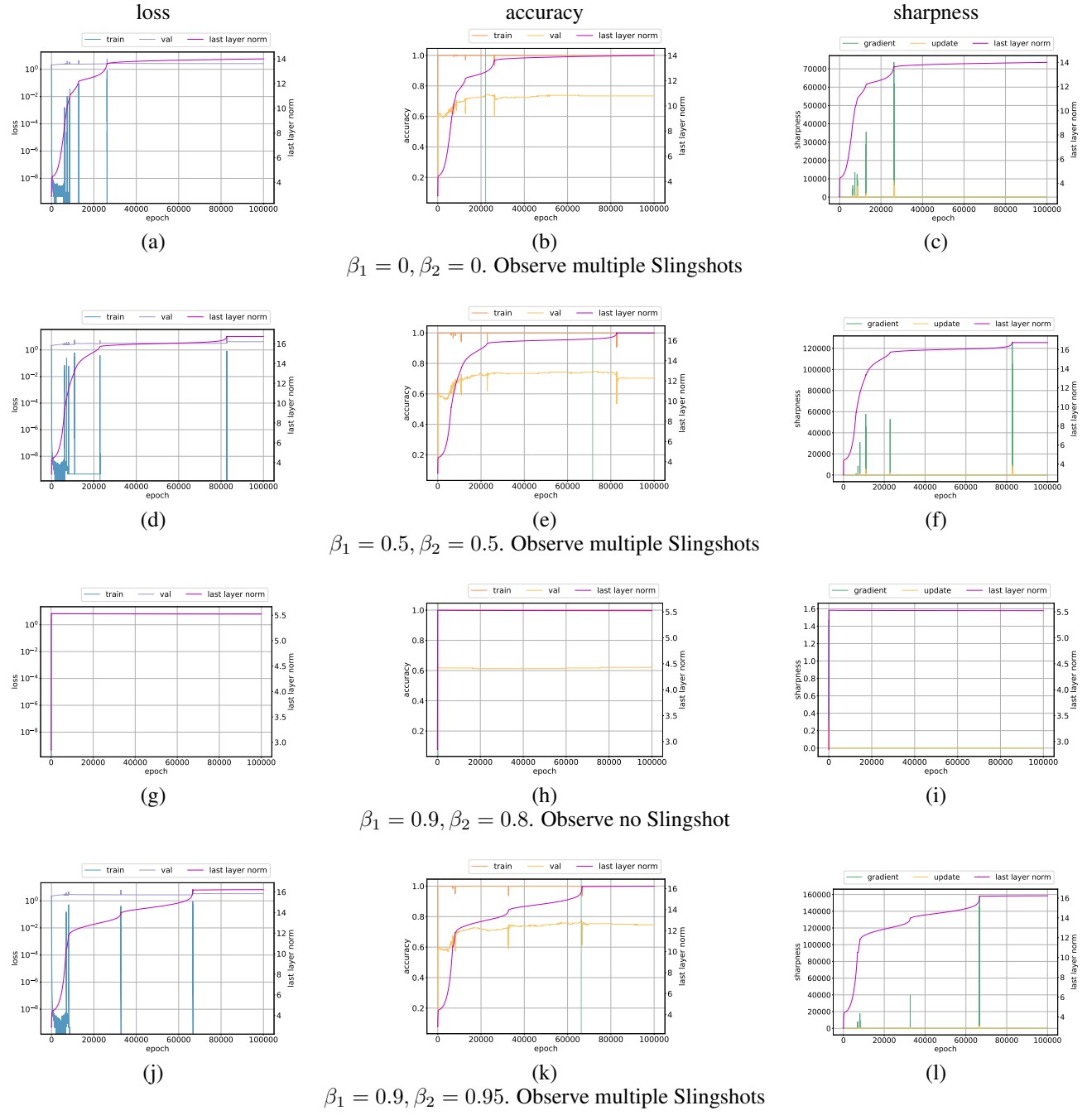

Figure 14: Varying $\beta_1, \beta_2$ in Adam on synthetic dataset. FCN is trained with Adam using learning rate $0.001$ and $\epsilon = 10^{-6}$. The validation accuracy of models that experience Slingshot reach their highest accuracy later in training.

199    $c = a \cdot b$ for $a, b \in S_5$

200    $c = a \cdot b \cdot a^{-1}$ for $a, b \in S_5$

201    $c = x \cdot b \cdot a$ for $a, b \in S_5$

202    $c = [a + b \pmod{p}$ if $a$ is even, otherwise $a * b \pmod{p}]$ for $0 \le a, b < p$

203    $c = [a + b \pmod{p}$ if $a$ is even, otherwise $a - b \pmod{p}]$ for $0 \le a, b < p$

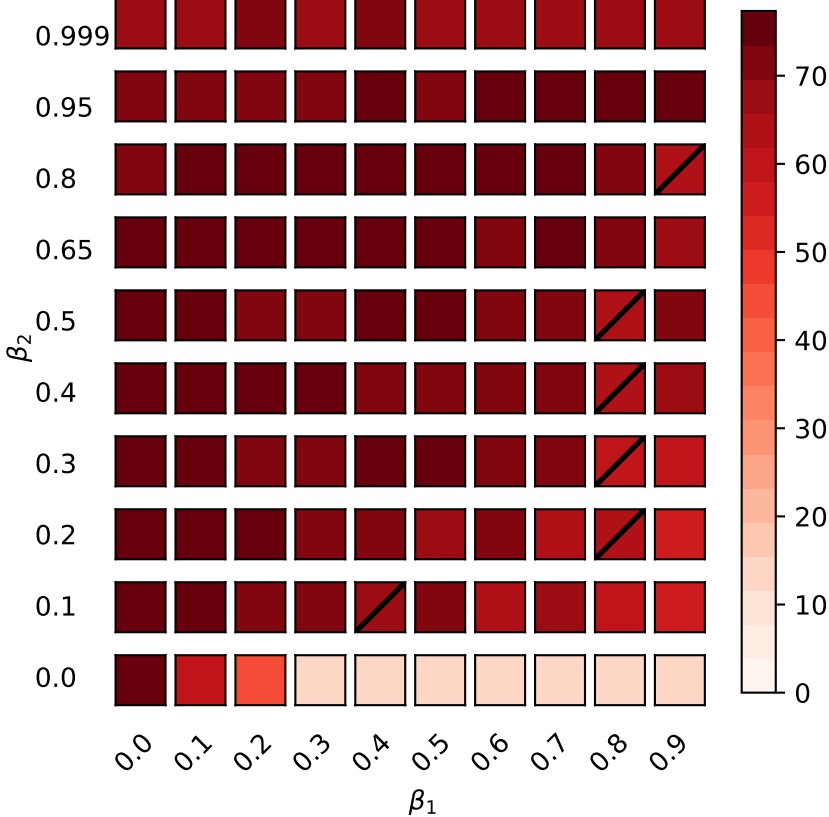

Figure 15: Extended analysis of $\beta_1, \beta_2$ in Adam on synthetic dataset. Plot shows the highest validation accuracy achieved with various values of $\beta_1, \beta_2$ with learning rate set to $0.001$ and $\epsilon = 10^{-6}$. Hyperparameters that do not induce Slingshot Effects are marked with a diagonal line in black. Models trained with $\beta_1 > 0.2$ and $\beta_2 = 0$ diverged during training due to instability. These trials have their validation accuracy set to chance level.

where $p = 97$ and with the dataset split in training and validation data. Each equation in the dataset is of the form $(a)(op)(b)(=)c$ where (x) represents the token used to represent x. We refer to Power et al. [12] for a detailed description of the datasets

## B.1 Analysis of Parameter Dynamics

A common observation is that intermediate representations tend to evolve beyond simple scale increase during phase transitions from norm growth to plateau. In order to empirically quantify this effect, we train the Transformer described in Appendix B with modular addition, multiplication and division datasets using Adam with learning rate set to $0.001$ and $\beta_1 = 0.9$ and $\beta_2 = 0.98$. We calculate the cosine distance between the representation and classification parameters from their initial values where the cosine distance is given by

$$d^{repr} = 1.0 - \frac{w_t^{repr}}{\|w_t^{repr}\|} \cdot \frac{w_0^{repr}}{\|w_0^{repr}\|}$$

$$d^{clf} = 1.0 - \frac{w_t^{clf}}{\|w_t^{clf}\|} \cdot \frac{w_0^{clf}}{\|w_0^{clf}\|}$$

where $d^{repr}$ ($d^{clf}$) denotes cosine distance for representation (respectively classification) parameters, $w_t^{repr}$ (resp. $w_t^{clf}$) denotes representation (resp. classification) parameters at time $t$ with $w_0^{repr}$ ($w_t^{clf}$)

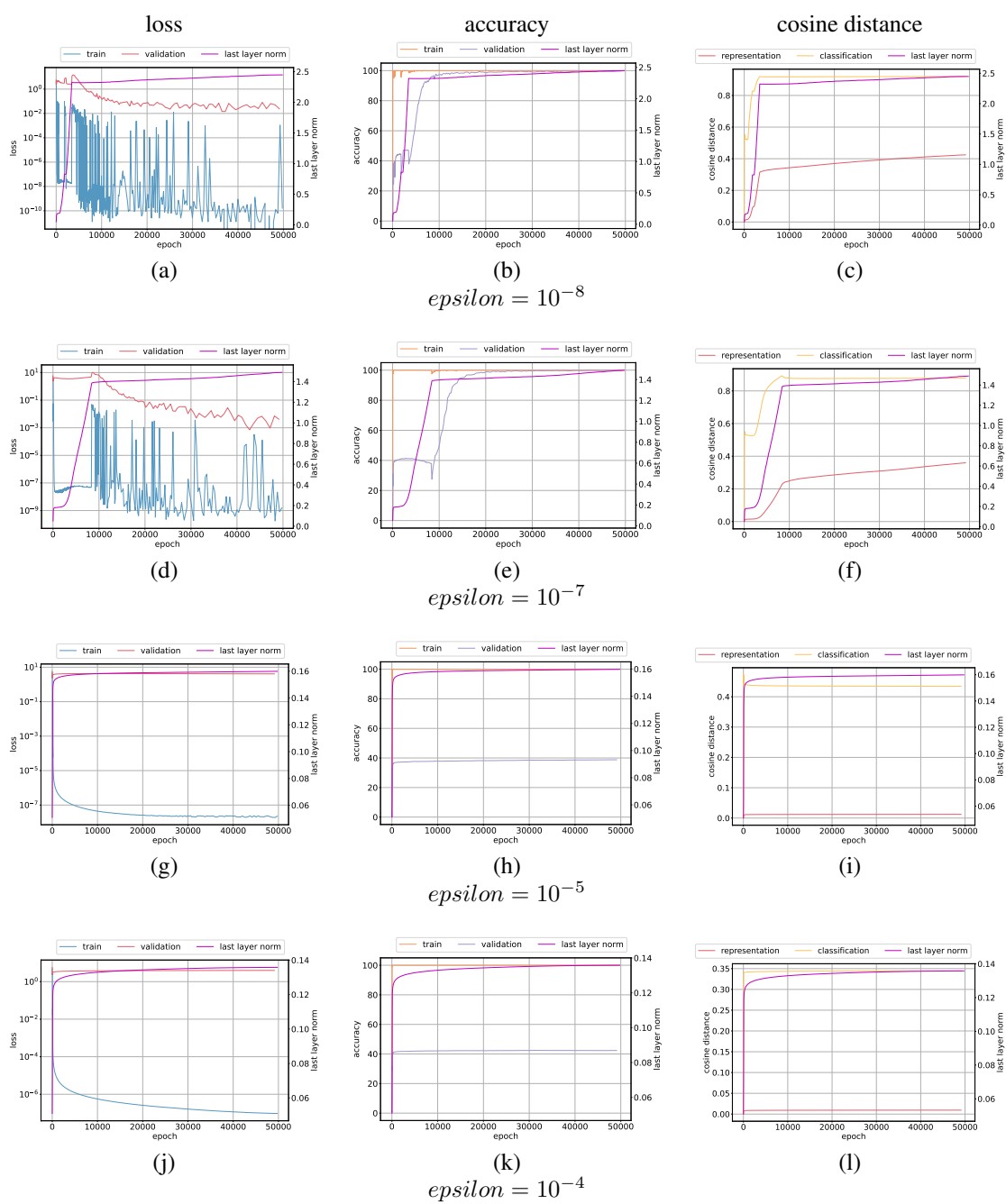

Figure 16: Cosine distance evolution for Transformer described in Appendix B trained on modular addition. Observe that the cosine distance from initialization increases with models that experience Slingshot Effects.

indicating the initial representation (resp. classification) parameters where the norm used above is the Euclidean norm.

Figure 16 shows the dynamics of the loss, accuracy and cosine distance recorded during training. We observe that the classification parameters move farther away from initialization faster than the representation parameters. More interestingly, we observe from Figure 16c and Figure 16f that the representation parameters travel farther from initialization for training runs that experience Slingshot. These trials use $\epsilon = 10^{-8}$ and $\epsilon = 10^{-7}$ and experience Slingshot Effects. In contrast, we see from Figure 16i and Figure 16l that the representation distance remains low for models trained with

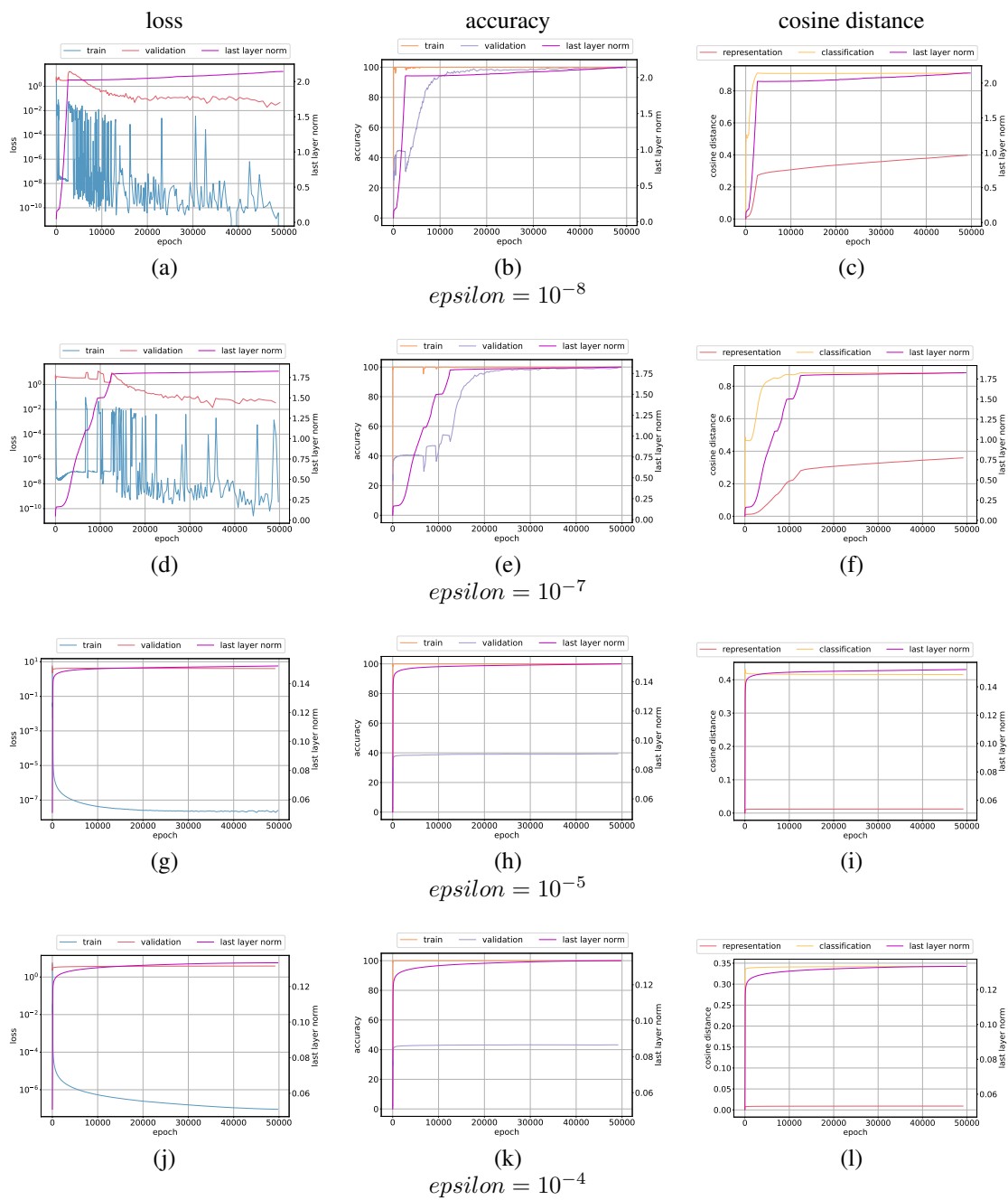

Figure 17: Cosine distance evolution for Transformer described in Appendix B trained on modular multiplication. Observe that the cosine distance from initialization increases with models that experience Slingshot Effects.

$\epsilon = 10^{-5}$ and $\epsilon = 10^{-4}$. The models trained with higher $\epsilon$ values do not experience Slingshot Effects. These results suggest that Slingshot may have a beneficial effect in moving the representation parameters away from initialization which eventually helps with model generalization. Figure 17 and Figure 18 show a similar trend for multiplication and division datasets respectively.

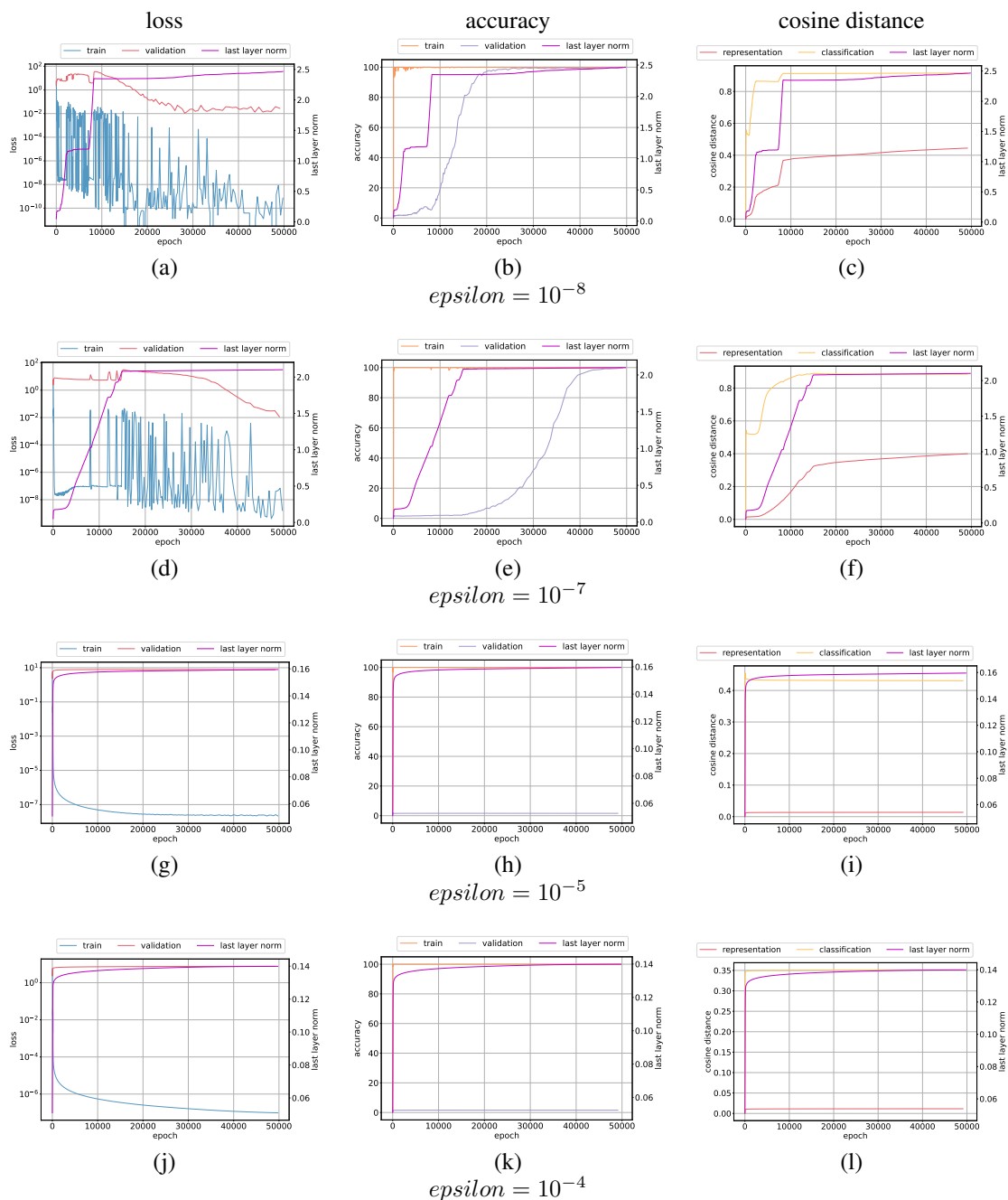

Figure 18: Cosine distance evolution for Transformer described in Appendix B trained on modular division. Observe that the cosine distance from initialization increases with models that experience Slingshot Effects.

## B.2 SGD Optimization

In this appendix, we show that Slingshot Effects are not seen during Transformer training with stochastic gradient descent (SGD) with momentum to support our claim in the main paper. To this end, we use train the Transformer described in in Appendix B on modular division dataset with a 50/50 train/validation split using SGD with momentum. We use a mini-batch size of $512$ which requires the optimizer to take 10 steps per epoch for dataset split described above. We set momentum to $0.9$ and use the following learning rates: $0.001$, $0.01$ and $0.1$ and run the optimizer for $1500000$

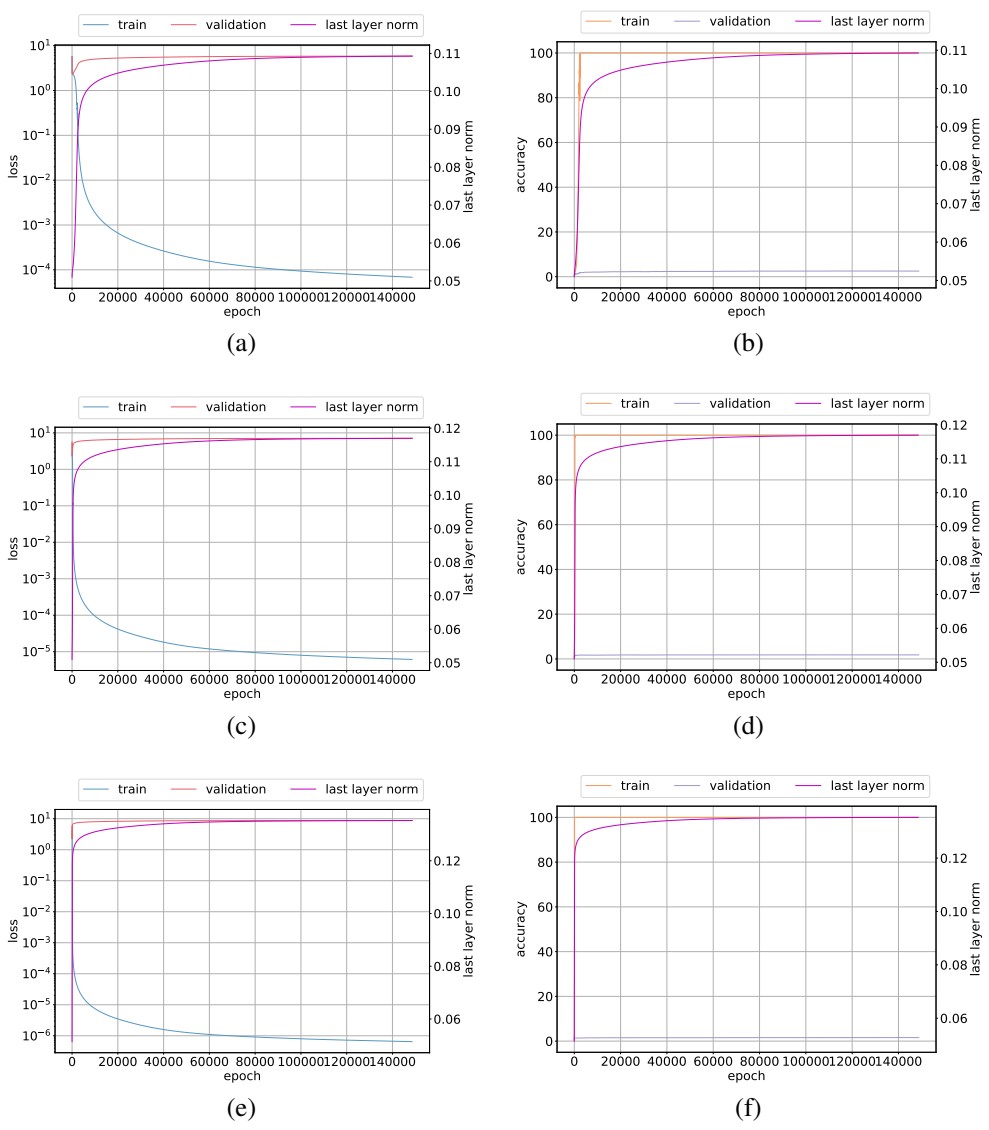

Figure 19: Optimizing a Transformer with SGD on modular division dataset: Norm growth vs (a), (c), (e) training and validation loss, (b), (d), (f) training and validation accuracy. Note the lack of Slingshot Effects, Grokking and generalization seen with Adam/AdamW optimizer.

steps. The number of steps used here is 3 times larger than the steps used to run Adam/AdamW in this work which is chosen to give SGD additional time to reach convergence. Figure 19 shows the usual loss and accuracy metrics calculated on training and validation data as well as the weight norm of the classifier layer. We observe that there is no evidence of Slingshot with SGD. Lastly, we do not see any evidence of Grokking or generalization with this setup as well.

## B.3 Slingshots with Additional Datasets

In this appendix, we provide evidence of Slingshot Effects on additional datasets from Power et al [12] Grokking work. The datasets are created by a subset of mathematical operations defined in Appendix B. Each operation can have multiple datasets that depends on the train/validation split ratio. We use the training setup described in B on 18 separate datasets. Figure 20 - Figure 37 shows the results the datasets described in this appendix. We observe Slingshot Effects and generalization

 with all 18 datasets. These results suggest the prevalence of Slingshot Effects when large models are
 trained with adaptive optimizers, specifically Adam [7].

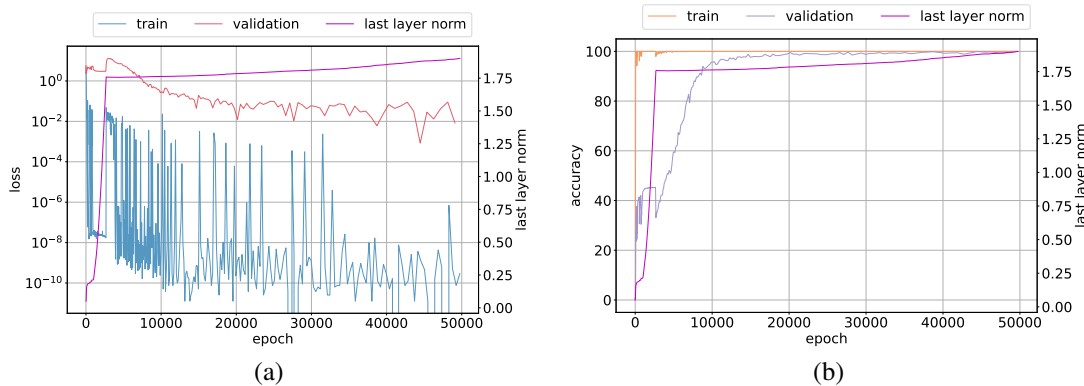

Figure 20: Addition dataset with 50/50 train/validation split. Training and validation (a) loss and (b) accuracy.

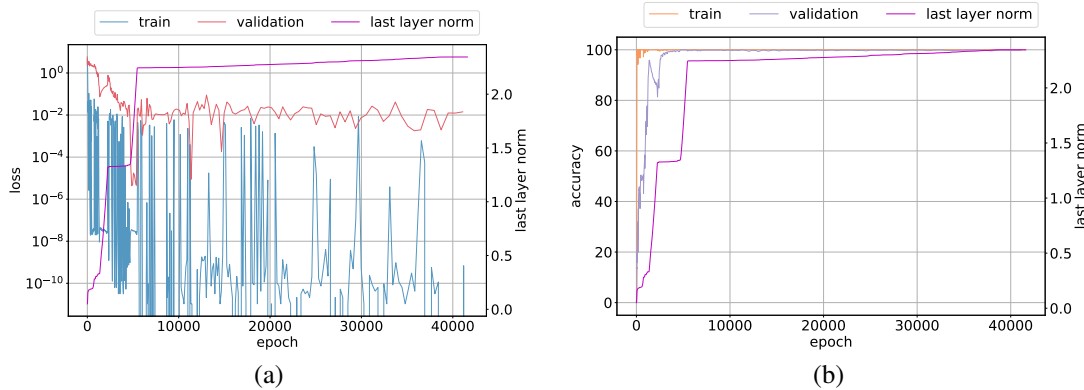

Figure 21: Addition dataset with 60/40 train/validation split. Training and validation (a) loss and (b) accuracy.

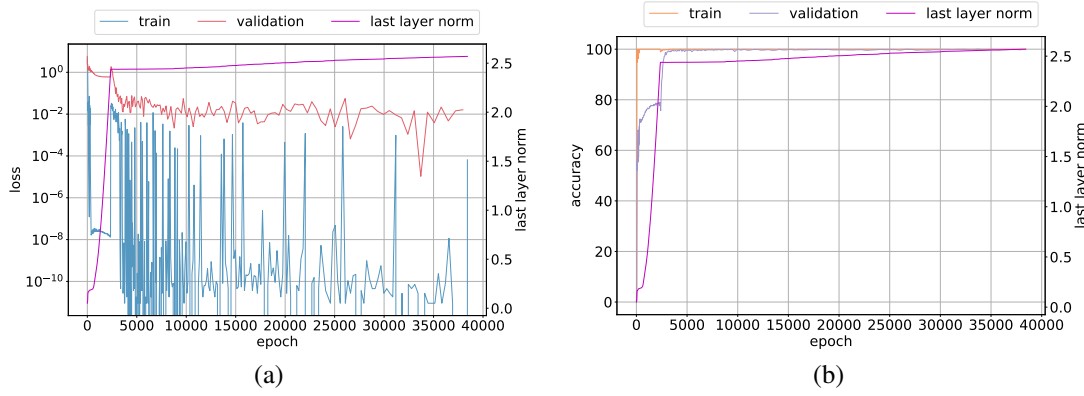

Figure 22: Addition dataset with 70/30 train/validation split. Training and validation (a) loss and (b) accuracy.

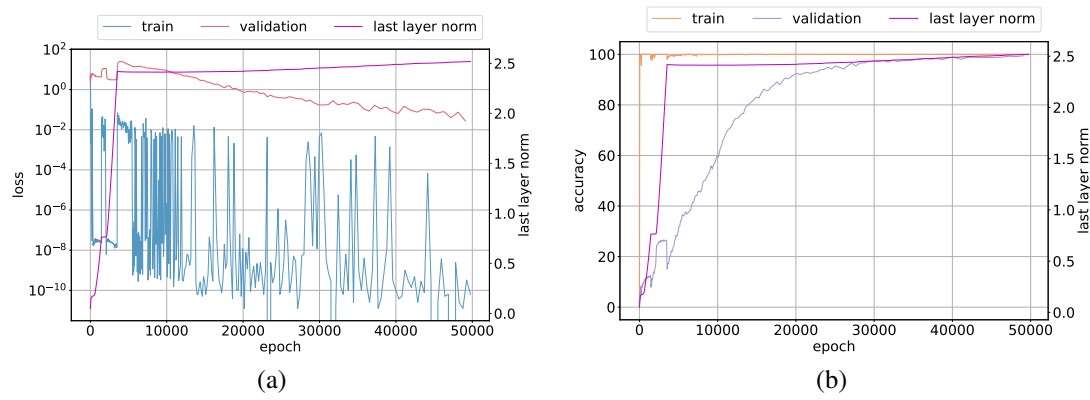

Figure 23: Cubepoly dataset with 50/50 train/validation split. Cubepoly operation is given by $(a^3 + b \pmod{p}$ for $0 \leq a, b < p)$. Training and validation (a) loss and (b) accuracy.

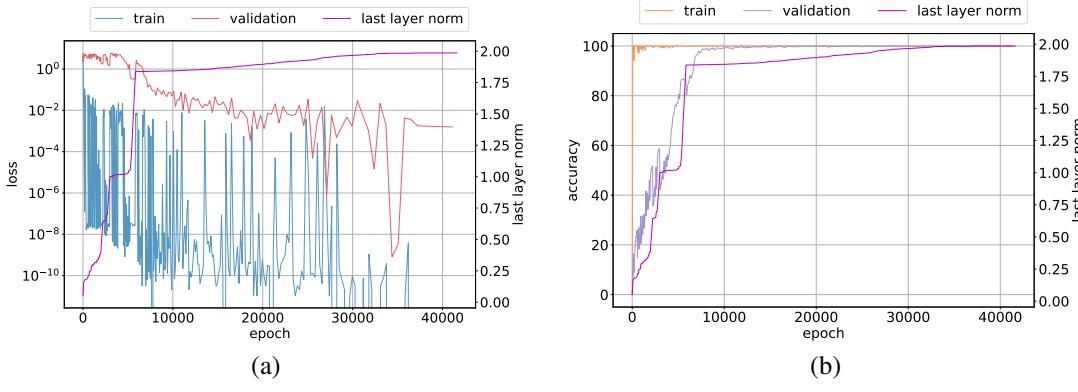

Figure 24: Cubepoly dataset with 60/40 train/validation split. Cubepoly operation is given by $(a^3 + b \pmod{p}$ for $0 \leq a, b < p)$. Training and validation (a) loss and (b) accuracy.

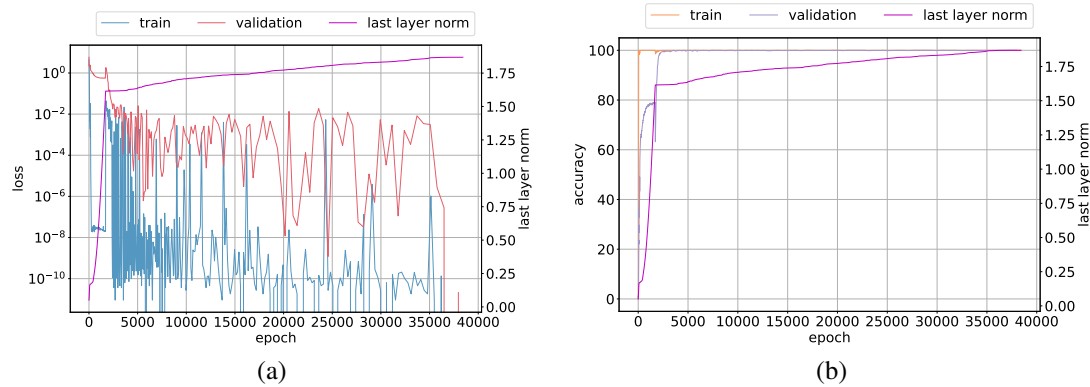

Figure 25: Cubepoly dataset with 70/30 train/validation split. Cubepoly operation is given by $(a^3 + b \pmod{p})$ for $0 \le a, b < p$). Training and validation (a) loss and (b) accuracy.

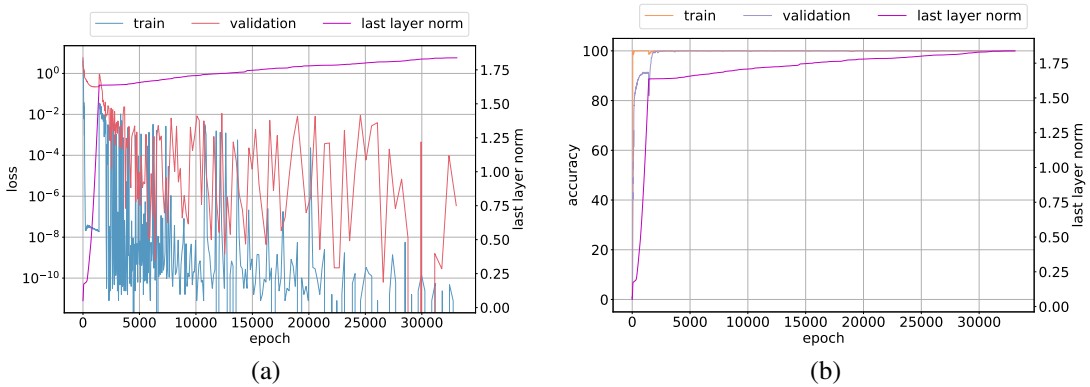

Figure 26: Cubepoly dataset with 80/20 train/validation split. Cubepoly operation is given by $(a^3 + b \pmod{p})$ for $0 \le a, b < p$). Training and validation (a) loss and (b) accuracy.

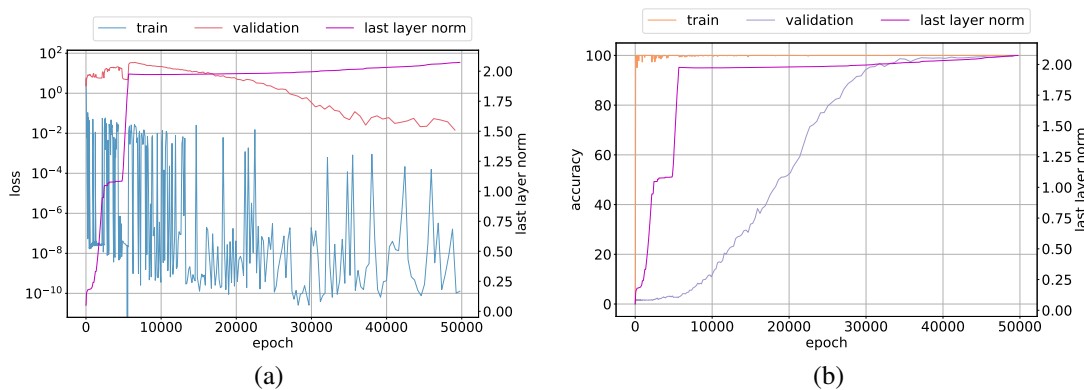

Figure 27: Division dataset with 50/50 train/validation split. Training and validation (a) loss and (b) accuracy.

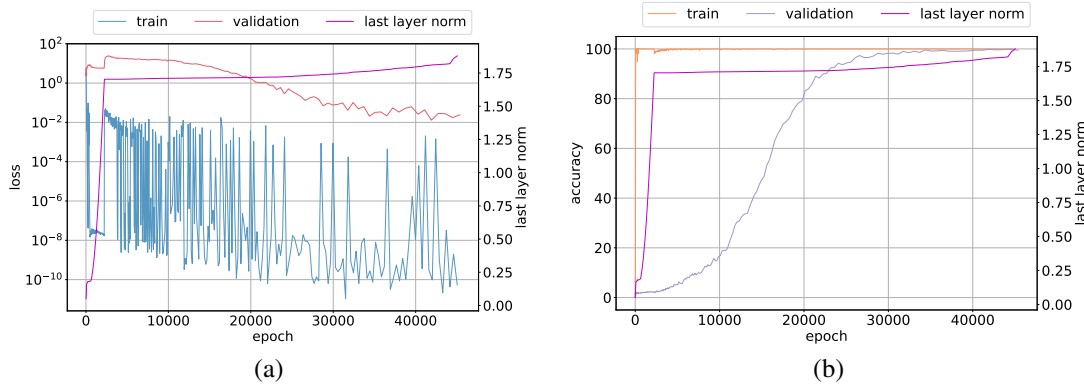

Figure 28: Division dataset with 60/40 train/validation split. Training and validation (a) loss and (b) accuracy.

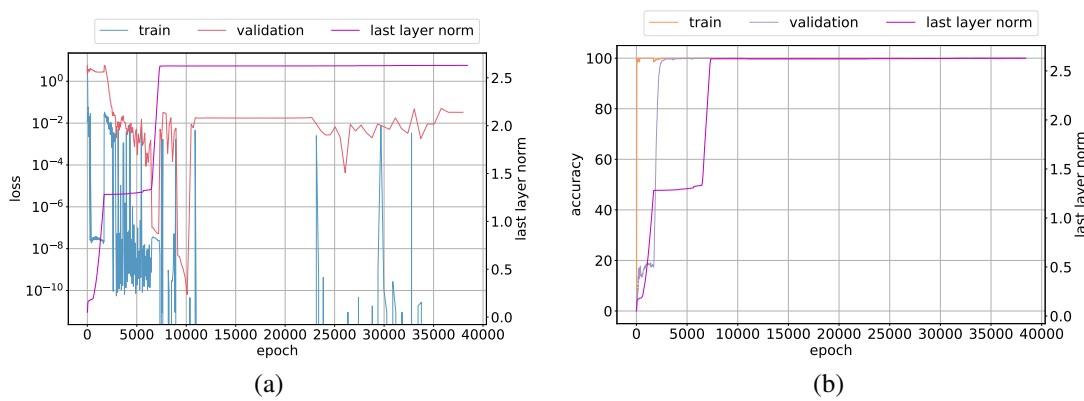

Figure 29: Division dataset with 70/30 train/validation split. Training and validation (a) loss and (b) accuracy.

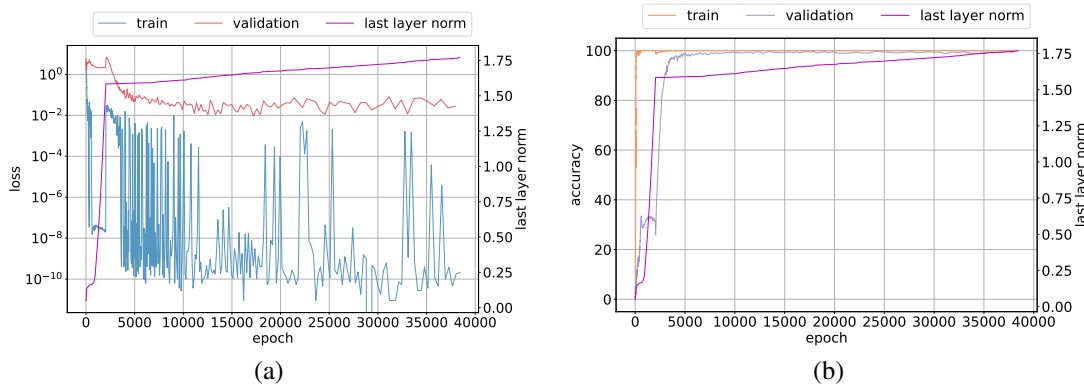

Figure 30: Even-add-odd-subtraction dataset with 70/30 train/validation split. Even-add-odd-subtraction operation is given by $[a + b \pmod{p}$ if $a$ is even, otherwise $a - b \pmod{p}]$ for $0 \le a, b < p$. Training and validation (a) loss and (b) accuracy.

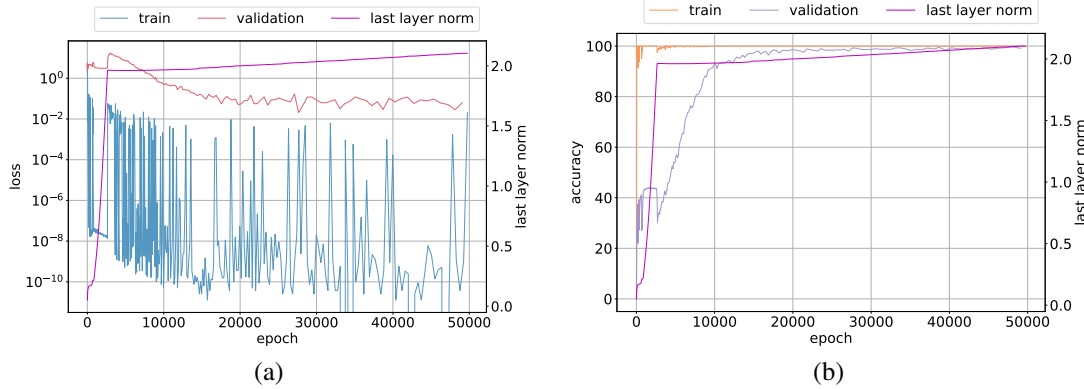

(a)                                                          (b)

Figure 31: Multiplication dataset with 50/50 train/validation split. Training and validation (a) loss and (b) accuracy.

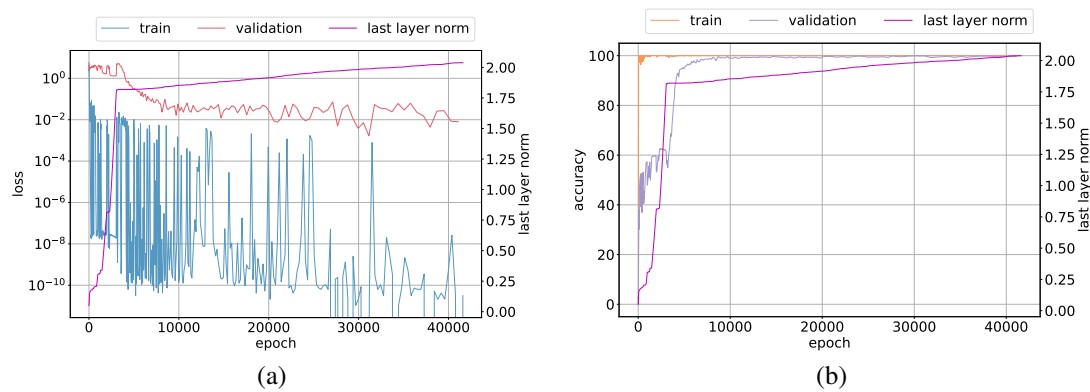

(a)                                                          (b)

Figure 32: Multiplication dataset with 60/40 train/validation split. Training and validation (a) loss and (b) accuracy.

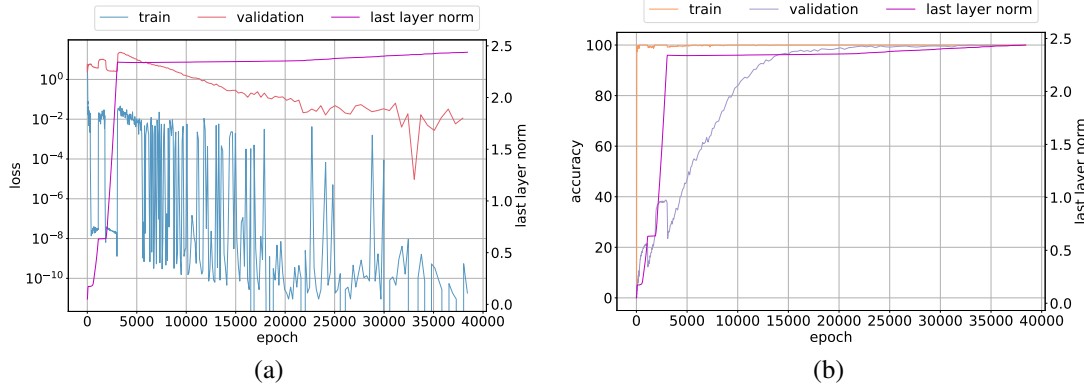

(a)                                                          (b)

Figure 33: Squarepoly dataset with 70/30 train/validation split. Squarepoly operation is given by $a^2 + b \pmod{p}$ for $0 \leq a, b < p$. Training and validation (a) loss and (b) accuracy.

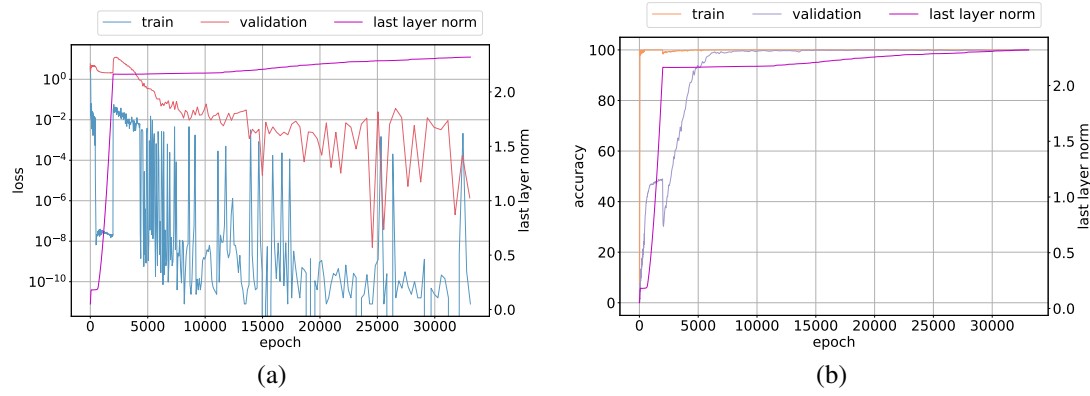

Figure 34: Squarepoly dataset with 80/20 train/validation split. Squarepoly operation is given by $a^2 + b \pmod{p}$ for $0 \le a, b < p$. Training and validation (a) loss and (b) accuracy.

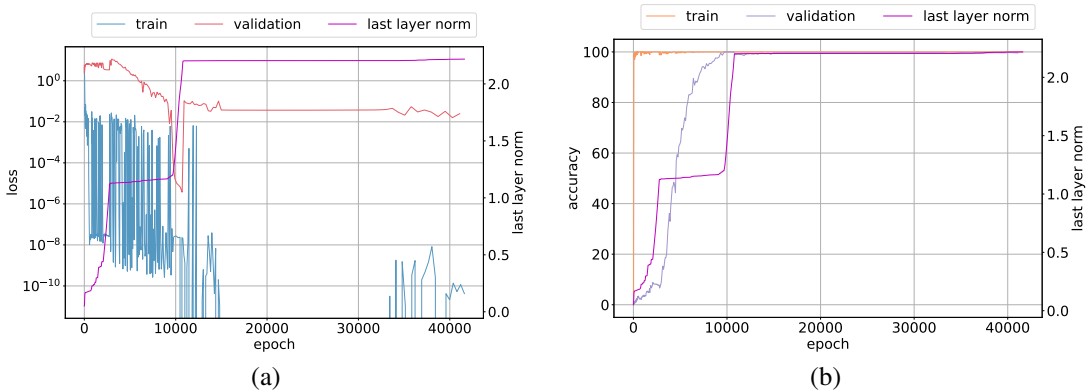

Figure 35: Subtraction dataset with 60/40 train/validation split. Training and validation (a) loss and (b) accuracy.

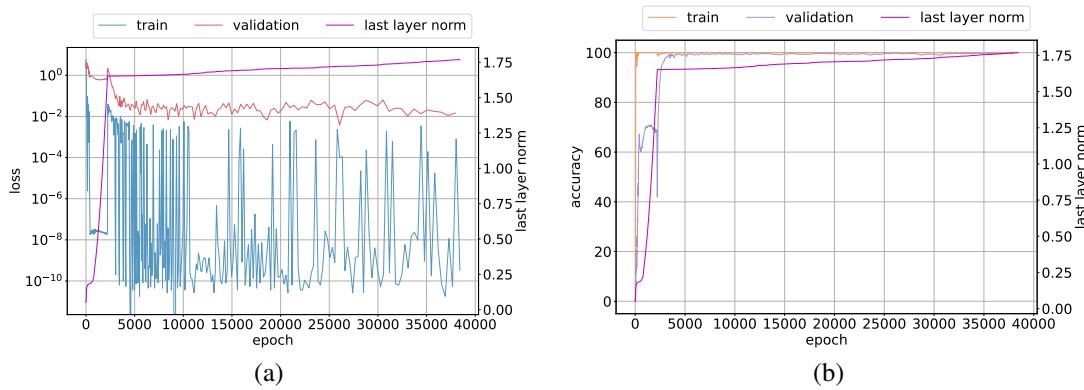

Figure 36: Subtraction dataset with 70/30 train/validation split. Training and validation (a) loss and (b) accuracy.

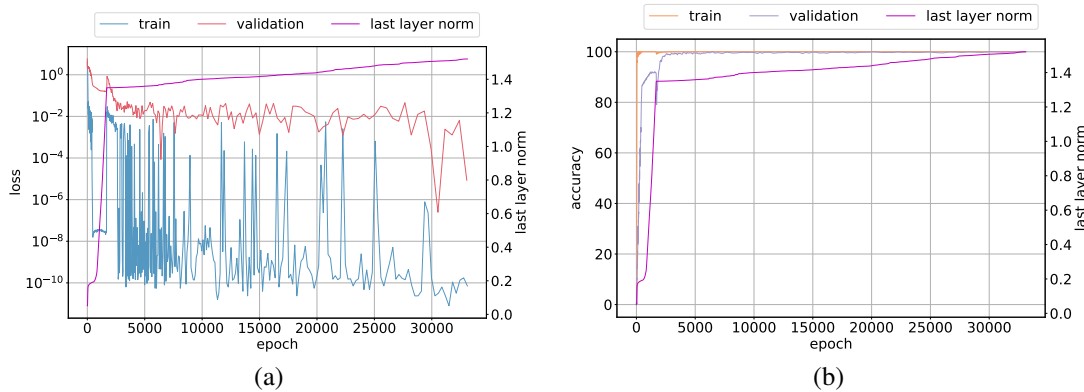

Figure 37: Subtraction dataset with 80/20 train/validation split. Training and validation (a) loss and (b) accuracy.

## C Controlling Instability Through Normalization and Norm Constraints

Training instability is the hallmark of the Slingshot Mechanism, yet as seen in previous sections, the Slingshot Effect typically results in improved performance, and Grokking. In this section, we explore whether it is possible to maintain stable training, without sacrificing performance. To this end, we explore how constraining and regularizing the weights of the network affect the Slingshot behaviour, and overall performance.

### C.1 Weight decay

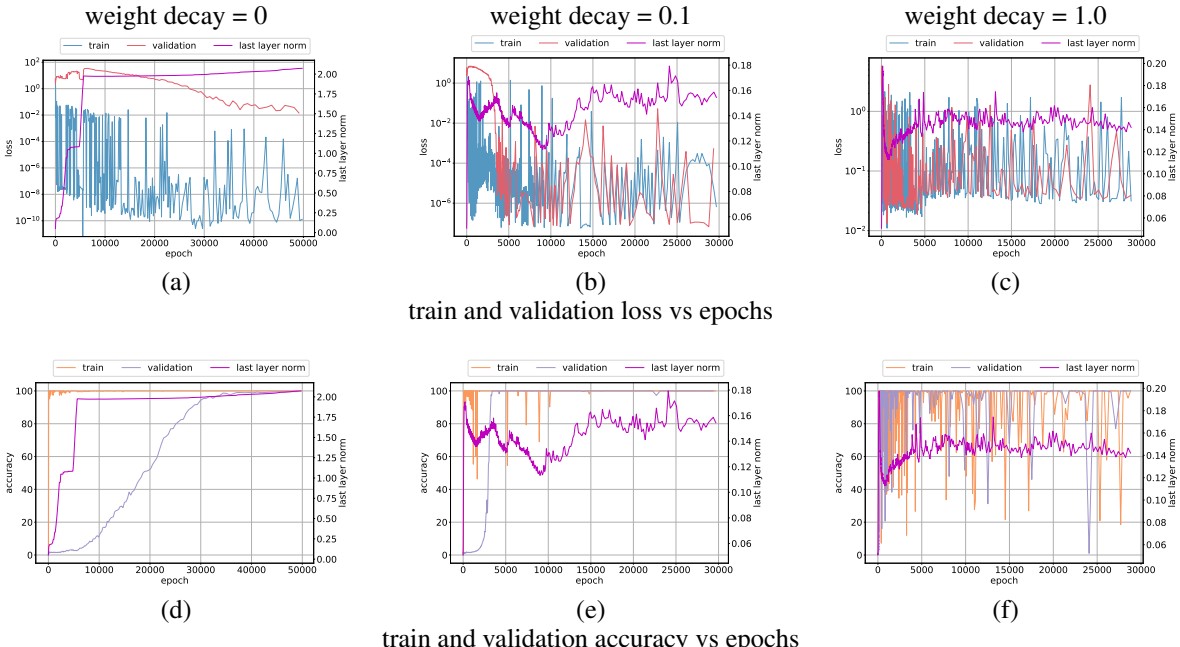

Figure 38: Division dataset: Norm behavior with different weight decay values. Training and validation loss vs epochs with weight decay (a) 0.0, (b) 0.1, (c) 1.0; Training and validation accuracy vs epochs shown in (d), (e) and (f). The evolution of classifier weight norm shows instability as increase in weight decay strength.

Weight decay is a commonly used regularization approach to improve the generalization performance of neural networks. Power et al. [12] show that weight decay has the largest positive effect on alleviating Grokking. Weight decay naturally controls the size of the parameters and consequently their norm growth. We study the effect of weight decay on stability of training Transformers with Grokking datasets in this section. We use weight decay values from $0, 0.1, 0.2, 0.4, 0.6, 0.8 and 1.0$ with AdamW [9] optimizer. Figure 38 shows the results for division dataset. We observe from Figure 38 that as weight decay strength increases, both Slingshot Effects and Grokking phenomenon disappear with the model reaching high validation accuracy quickly as seen in Figure 38e and Figure 38f. However, we observe that the model experiences instability as can been seen with the loss plots in Figure 38b and Figure 38c or the accuracy plots in Figure 38e and Figure 38f. A similar trend is observed for addition and multiplication datasets in Figure 39 and Figure 40 respectively.

The results shown above indicate that Slingshot may not be the only way to achieve good generalization. Both Slingshot and weight decay prevent the norms from growing unbounded and achieve high validation accuracy as seen in plots described above. While weight decay shows different weight norm dynamics, this regularization does not decrease training instability. These results suggest the need for alternative approaches to improve training stability.

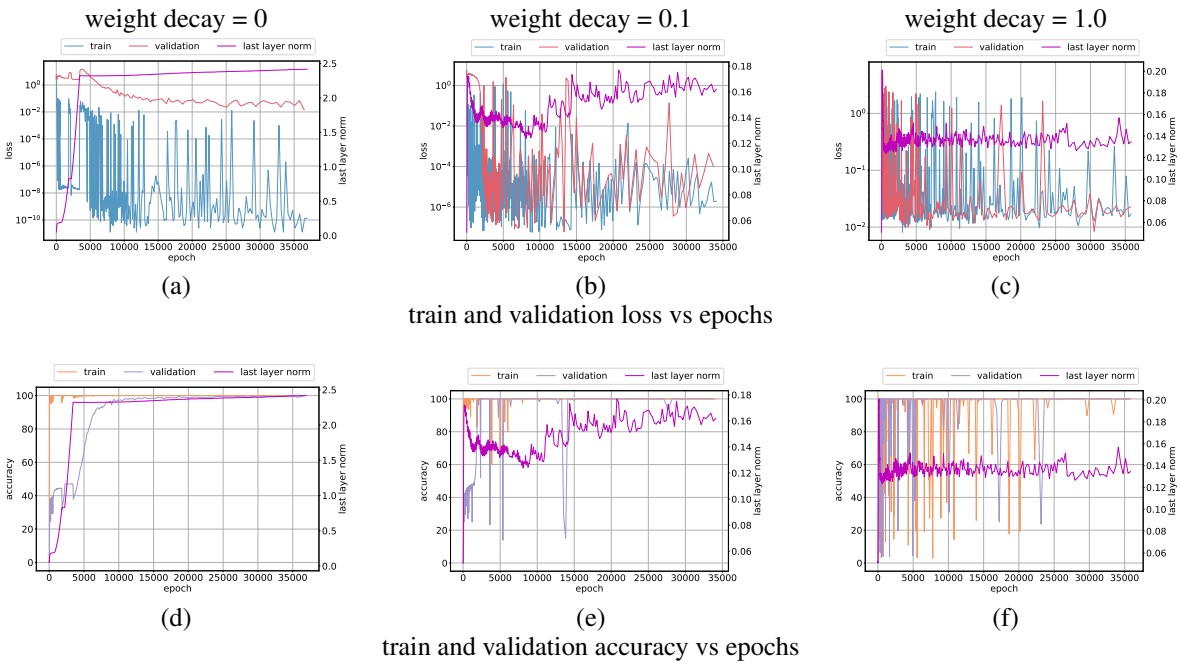

Figure 39: Addition dataset: Norm behavior with different weight decay values. Training and validation loss vs epochs with weight decay (a) 0.0, (b) 0.1, (c) 1.0; Training and validation accuracy vs epochs shown in (d), (e) and (f). The evolution of classifier weight norm shows instability as increase in weight decay strength.

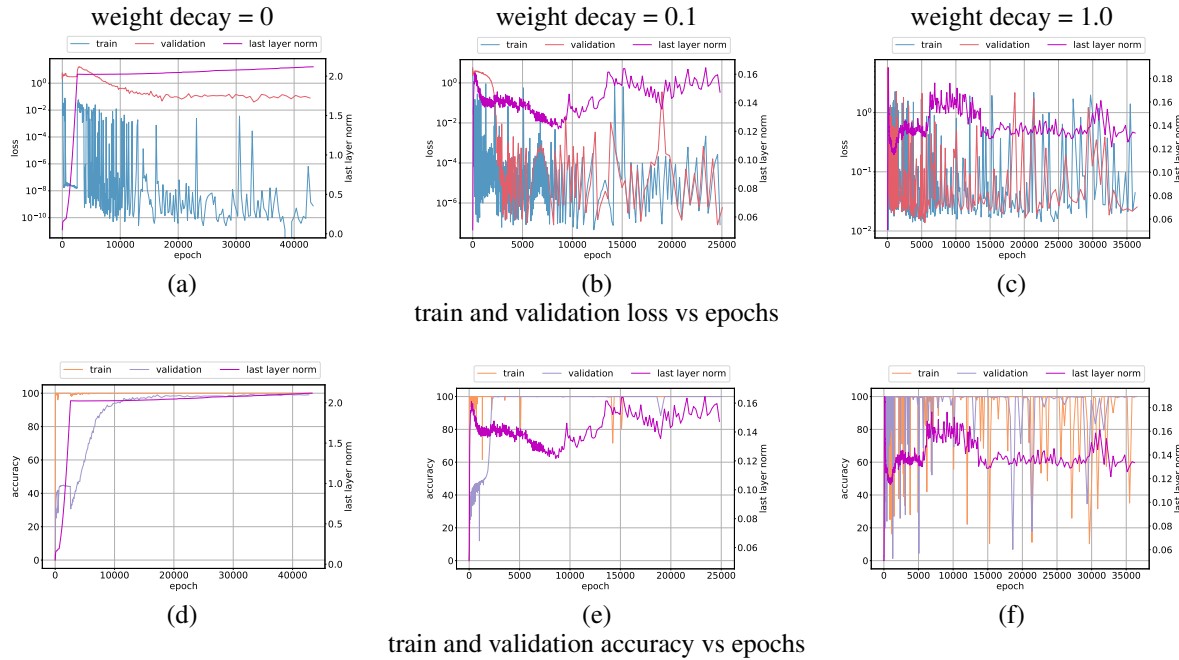

Figure 40: Multiplication dataset: Norm behavior with different weight decay values. Training and validation loss vs epochs with weight decay (a) 0.0, (b) 0.1, (c) 1.0; Training and validation accuracy vs epochs shown in (d), (e) and (f). The evolution of classifier weight norm shows instability as increase in weight decay strength.

## C.2 Features and parameter normalization

A second approach that we use to explicitly control weights and feature norm is by normalizing the features and weights via the following scheme: $w = \frac{w}{\|w\|}, f(x) = \frac{f(x)}{\|f(x)\|}$, where $w$ and $f(x)$ are the weights and inputs to the classification layer respectively, the norm used above is the $L_2$ norm, and $x$ is the input to the neural network. We take the cosine similarity of the normalized weights and features and divide this value by a temperature value that we treat as a hyperparameter in these experiments. The operation is given by: $y = \frac{w \cdot f(x)}{\tau}$ where $\tau$ represents the temperature hyperparameter. We use temperature values from $0.1, 0.25, 0.5, 0.75, 1.0$ for these experiments.

Figure 41 shows the results of Transformer training on division dataset described in Appendix B that is split evenly into train and validation sets. We observe that the model displays training instability evidenced by norm behavior and also loss behavior in Figure 41a at lower temperature values. We observe that $\tau = 0.25$ provides a good compromise between fitting training data while showing no training instability as seen in Figure 41b. This hyperparameter value also results in Grokking as validation accuracy improves late in training as can be seen from Figure 41e. These together suggest that bounding weights and features norm helps stabilize training without sacrificing training performance.

We validate the normalization scheme with two additional datasets namely multiplication and division from Appendix B. Figure 42 shows the results for training Transformers with multiplication dataset that is split evenly into train and validation sets. We observe from Figure 42 that a proper temperature value can stabilize training and with some tuning can provide a compromise between training stability and generalization. Specifically, $\tau = 0.25$ allows the model to fit the training data and reach almost perfect validation accuracy as seen from Figure 42b and Figure 42e.

Finally, we repeat the above experiments with subtraction dataset and show the results in Figure 43. This dataset shows that while a properly tuned temperature can help the model achieve almost perfect generalization, training instability shows up very late in optimization. This observation can be seen from Figure 43b and Figure 43d. This result suggests that more work remains to be done with understanding and stabilizing the training behavior of large neural networks.

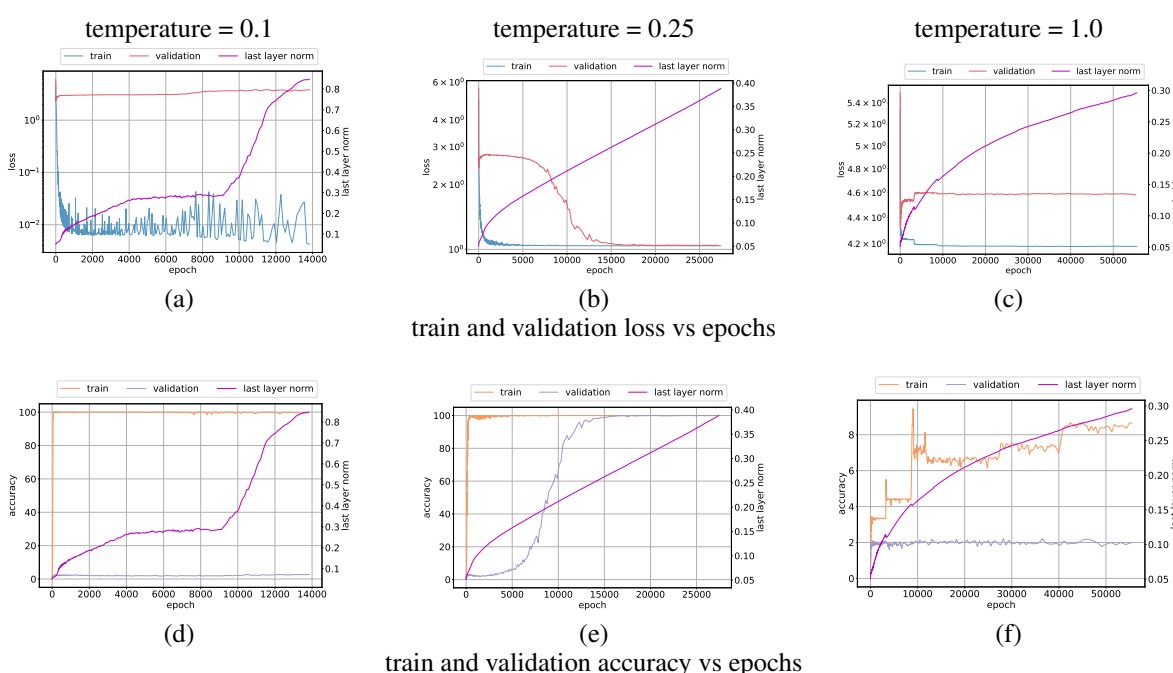

Figure 41: Division dataset: Features and parameters normalization. Observe that a smaller temperature allows the model to fit the data better but experiences training instability. Temperature = 0.25 allows the model to fit and achieve high validation accuracy without suffering training instability.

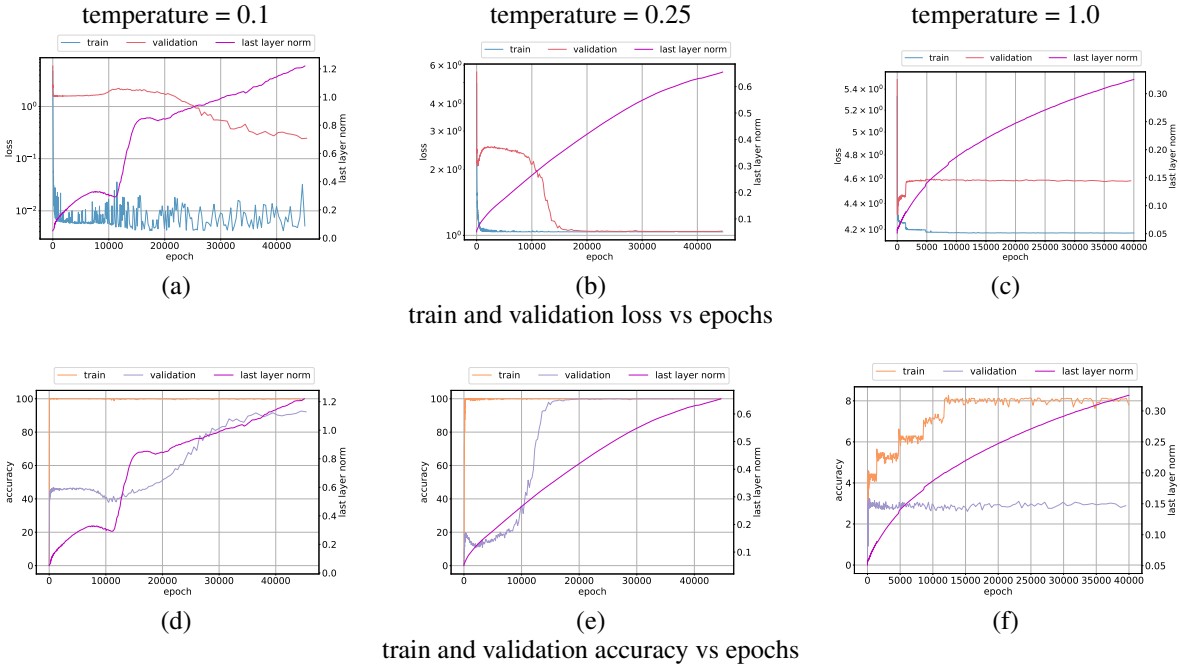

Figure 42: Multiplication dataset: Features and parameters normalization. Observe that a smaller temperature allows the model to fit the data better but experiences training instability. Temperature = 0.25 allows the model to fit and achieve high validation accuracy without suffering training instability.

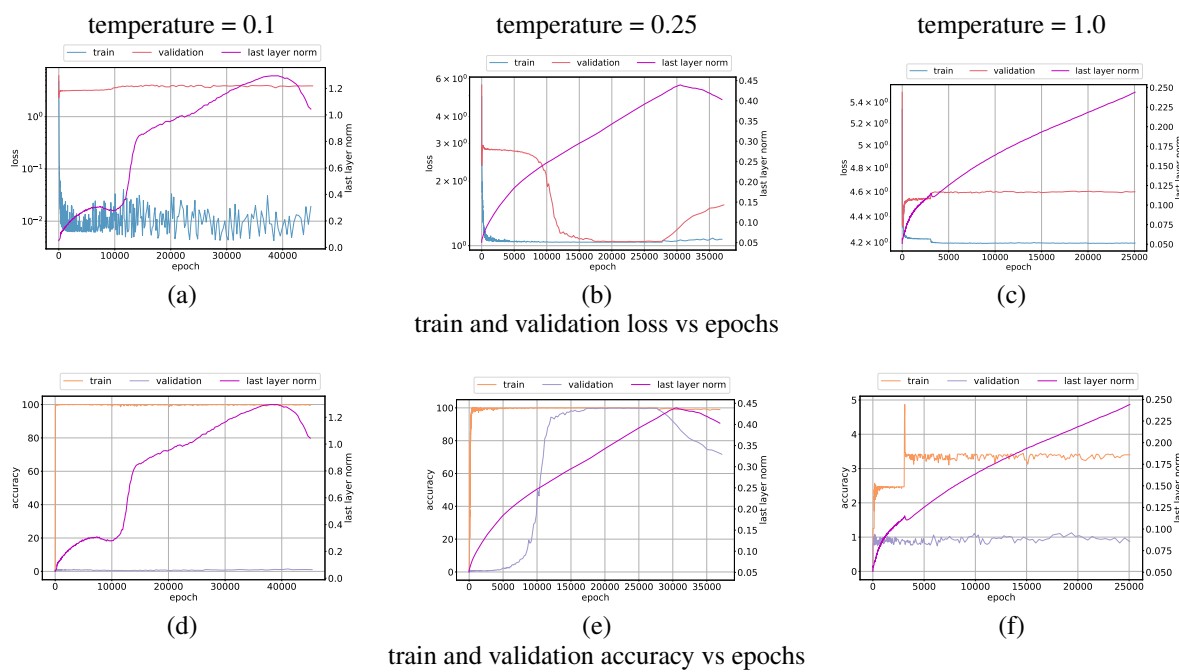

Figure 43: Subtraction dataset: Features and parameters normalization. Observe that a smaller temperature allows the model to fit the data better but experiences training instability. Temperature = 0.25 allows the model to fit and achieve high validation accuracy. However, we observe training instability as can seen with weight norm dynamics.