# OpenReview forum: "The Slingshot Mechanism: An Empirical Study of Adaptive Optimizers and the \emph{Grokking Phenomenon}"
_NeurIPS.cc/2022/Conference — NeurIPS 2022 Submitted_

### Official Review · Reviewer_8hZg · 2022-07-12

**Rating:** 6
**Confidence:** 3
**Soundness:** 3 good
**Presentation:** 3 good
**Contribution:** 2 fair

**Summary:**

The authors describe a "slingshot" effect present in the use of Adam (and cousins). This purportedly cyclic effect can be observed when the norm of the final layer sharply increases, often accompanied by an increase in train loss. The authors offer the slingshot effect as a possible explanation for the ability of Adam to induce generalization even without explicit regularization.

**Questions:**

1. Is there a reason validation error/loss is not depicted in Fig 1? Given the emphasis in this paper on generalization, it seems odd not to depict in the headline figure.
2. Figure 2(c) looks either incorrect or misgenerated—the validation loss should not be zero at the beginning of training, unless I am missing something.
3. Do the authors have any explanation for why Figure 3(d) doesn't show a high-curvature update on the second plateau?
4. Can the authors elaborate more on the implications of the larger $\epsilon$ result? To me, larger $\epsilon$ ruining the grokking effect sort of implies that non-adaptive methods (e.g. plain GD) don't benefit from slingshotting either (which is exactly what the authors found empirically). But it would still be good to explicitly connect the dots here.
5. I am also left wondering about the result on the paragraph starting on line 243. It's cool that the authors discovered it but what is it supposed to tell me? What about low-dim data makes the slingshot effect less likely to occur?
6. Finally, is it fair to call this effect cyclic when most empirical examples only have two cycles (with a few having 3)? Is there a setup where we can see $n$ distinct plateaus and sharp norm increases for some $n >> 3$?

**Limitations:**

The authors do a good job not overclaiming, and they explicitly enumerate a number of other regimes that achieve the results despite lack of "slingshot". Certain items are left unaddressed (see "Questions").

Minor
----

- Line 84: missing author name
- Line 112: what is "Adam without momentum" and how is it different from RMSProp?
- Line 145: "layers" $\rightarrow$ "layer"
- Line 170: "has" $\rightarrow$ "have"
- Line 245: "choices" $\rightarrow$ "choice"
- Line 254: "it's" $\rightarrow$ "its"

**Strengths And Weaknesses:**

This is overall a nicely executed empirical paper that reveals a moderately interesting phenomenon. A cyclic regime of generalization would be quite remarkable and this paper offers preliminary evidence that supports the authors' hypothesis. It is also refreshing that the authors did not attempt to handwave nonsense poorly-motivated math into the paper, which is unfortunately a severe problem even in 2022.

My main concerns are the following:

- Significance: Given that this effect explains generalization in unregularized optimization, the overall impact to practical deep learning is somewhat muted.
- Strength of evidence: the results in the paper leave me with certain questions (most notably #6 below) as to whether the authors have sufficiently validated their hypothesized effect.
- Implications: the paper enumerates a number of interesting facts (e.g. dependence on $\epsilon$) but doesn't really dive deeper beyond some of those facts, leaving the reader wondering as to why those facts were presented.

Overall, I enjoyed reading this paper. I don't think it's super noteworthy in terms of immediate impact, but it is (as far as I can tell) a novel discovery that may serve as a waypoint on our path to better understanding of oddball generalization phenomena in deep learning.

---

> ### Author Response · Authors · 2022-08-02
> **Response to Reviewer 8hZg**
>
> We thank the reviewer for their review and comments, and their support for our paper. Please see our response bellow:
>
> **Significance**
> - We agree that the practical implication to unregularized training is not clear, however there is much value to be had in understanding the implicit biases in unregularized settings.  Understanding deep learning requires disentangling multiple confounding factors, and we argue it is especially hard to study a phenomena empirically or theoretically without isolating it from other confounders. Many such empirical investigations in less than “typical” settings (for instance, full batch gradient descent - [1]) have been carried out in the field, resulting in profound theoretical understanding later on.
>
> **Strength of evidence + Implications:** Please see our answers to the raised questions below.
>
> **Questions**
>
> 1. Validation curves are missing in figure (1) because these are taken in a setting with only 200 training samples on CIFAR10 (so generalization is not the focus), a setting designed to illustrate the optimization phenomena which is the Slingshot Effect. Please see experiments on the algorithmic datasets, as well as our newly added experiments on learning subset parities (appendix A.6) for a more dramatic illustration on the apparent effect of Slingshots on generalization.
> 2. We thank the reviewer for catching the problem with Figure 2(c). This figure is indeed misgenerated. We have fixed this plot in the revised version of our paper.
> 3. We thank the reviewer for identifying and raising this issue. We have spotted an alignment bug in Figure 3(d). As a consequence, we have rerun the experiment and regenerated the figure.
> 4. We believe the reviewers intuition regarding the effect of epsilon is correct here. A high value for epsilon would essentially mean that the adaptive updates are closer to SGD updates when the gradient magnitude is small - exactly when you expect the Slingshots to occur. We will make this point clear in future versions.
> 5. This property is indeed under-explored in the paper, and shall be removed from future versions to prevent confusion. We speculate it is harder to observe Slingshot effects when the data dimension is much smaller than the number of training samples since that would limit the ability of a network to overfit, but this is indeed just speculation.
> 6. The number of cycles do seem to depend on the data, however we show experiments where the number of cycles is clearly larger than 3 (see figure 2, 3, 5 and 7 in the appendix).
>
> Minor
>
> * Line 112: what is "Adam without momentum" and how is it different from RMSProp?
>     Adam without momentum contains an additional bias correction term in the denominator that is not present in RMSProp in Wang et al. [2] (specifically Section 3.1 in Wang et al.).
>
> We thank the reviewer for carefully reading our paper and bringing these typographical errors to our notice. We have fixed these mistakes in the updated version of our paper available for review. We will also carefully review and fix any remaining typographical errors is the final version of our paper.
>
> ### References
> - [1]  Cohen et al - “Gradient descent on neural networks typically occurs at the edge of stability”
> - [2] Wang, Bohan, et al. "The implicit bias for adaptive optimization algorithms on homogeneous neural networks."

---

> > ### Comment · Reviewer_8hZg · 2022-08-02
> > **Thank you**
> >
> > Thanks to the authors for their comprehensive response. I still maintain my statement of moderate significance but the authors have addressed many of my subsidiary concerns. Therefore, I weakly recommend acceptance and edit my score from 5 to 6.

---

### Official Review · Reviewer_KF2y · 2022-07-12

**Rating:** 5
**Confidence:** 4
**Soundness:** 2 fair
**Presentation:** 3 good
**Contribution:** 2 fair

**Summary:**

The paper studies the grokking phenomenon introduced by Power et al [1], which is a curious phenomenon of delayed generalization. This paper describes a characteristic of training termed the "slingshot mechanism", that is found in a number of tasks and optimization settings. The slingshot mechanism refers to periods of instability found in adaptive optimizers in the late stages of training. These periods of instability are accompanied by norm increases in the classification layer, as well as an improvement in generalization performance for the grokking tasks introduced in [1]. The paper studies this phenomenon and finds that it occurs in a number of different tasks more general than those in [1], and find that it is controlled by the epsilon parameter in adaptive optimizers.

**Questions:**

- In Figure 3(d), why do we not see a corresponding spike in sharpness at the onset of the second plateau? Based on the hypothesis that the weights are flung to a new region in parameter space when the curvature grows too large, I would expect to see the curvature spike at each transition.

- Figure 5, what is the accuracy being plotted? Is it training accuracy or test accuracy? If test accuracy, what is the training accuracy of these models? Do all the runs achieve 100% training accuracy? It is unclear whether correlation with better generalization comes from the appearance of slingshots or from the bad hyperparameter choices required to remove slingshot behavior from the training trajectory.

- Dataset size is a crucial parameter in determining whether grokking / generalization occurs in [1]. How does the correlation between generalization vs slingshot behavior change when models are trained with differing data size? Does slingshots exist regardless of whether the model generalizes?

**Limitations:**

Beyond the weaknesses I listed, the authors were good at addressing several limitations of this work.

**Strengths And Weaknesses:**

### Strengths

- The paper has a good and comprehensive discussion of related work.
- The paper characterizes an interesting phenomenon that is sometimes found in the optimization trajectory of adaptive optimizers.
- Interesting observations regarding the effect of epsilon on the slingshot behavior.

### Weaknesses

- There is no explanation, empirical or otherwise, for how the slingshot mechanism may relate to generalization. Understanding the delayed generalization is an important aspect of why the grokking phenomenon in [1] is interesting to the community. I believe that in the current state, without a deeper understanding, this work may be more appropriate for a workshop submission.
- The paper claims that the slingshot mechansim is a general phenomenon, however on CIFAR10 the phenomenon was mostly observed when training with extremely small training data (e.g. 200 examples), and no correlation to better generalization was shown. There is one experiment in the Appendix where the full CIFAR10 dataset is used in training a ViT model, but this model only achieves a final test accuracy of <80% and it's not clear how much test accuracy is improved due to the slingshot mechanism due to a single run and increasing test loss. This calls into question how relevant or general the identified behavior is to modern day neural networks.

---

> ### Author Response · Authors · 2022-08-02
> **Response to Reviewer KF2y**
>
> We thank the reviewer for their review and comments. Please see our response bellow:
>
> **There is no explanation, empirical or otherwise, for how the Slingshot mechanism may relate to generalization.**
> - This is a valid point raised by the reviewer. We view the Slingshot effect as an optimization anomaly, and its precise connection to generalization is at this point unclear. However, due to the generality of the phenomena and the popularity of adaptive optimizers, we feel our paper would provide valuable insight to the community (for more on this see our comments bellow). We also provide additional compelling evidence that Slingshot effect at least correlates well with generalization in a study on subset parity described below, which further convinces us these effects will be of interest. Finally, we argue that providing adequate explanations of interesting empirical phenomena should not pose a barrier to publication in the field, which could result in limiting exposure to important findings which are beyond current understanding.
>
> **On the generality of the Slingshot effect**
> - We view the Slingshot as an optimization anomaly rather than a generalization mechanism. As such, the Slingshot effect is general, and can be clearly seen in a variety of models ranging from a 2-layer MLP all the way to a transformer. A prominent staple of the effect however is that its onset happens at very late stages of training (long after training accuracy reaches 100%), which might hide its existence from practitioners.
> Training a large model on a large dataset long past the point of 100% training accuracy is naturally computationally demanding, therefore we focus on small datasets to more easily observe what happens in these regimes, which also aligns well with the setting put forth by [1]. We further argue that the small data/large training time regime is an interesting and relevant one in practice since in a variety of practical settings (especially in industry) a large dataset is typically expensive and hard to come by.
>
> **Performance on CIFAR10**
> - When running on extremely small datasets (200 samples), we do not expect to see any meaningful performance benefits on held-out data, and the experiment was strictly meant as a convenient testbed to observe the Slingshot effect in the optimization process, and to check that the effect also happens reliably in a small data version in a different (vision) domain. Note that in the full CIFAR experiment, small bumps in performance can be seen at (some of) the transitions.
> We further argue that state of the art performance is not the appropriate testbed when investigating a fundamental mechanism such as the Slingshot effect, which may be obscured by numerous hacks, tricks and hyperparameter optimization required to achieve SOTA results.
>
> **More on generalization -** As further experimental evidence to how the Slingshot effect is conducive to generalization, we have updated the appendix with more experiments. Specifically, we borrow the setting from [2], and use the k subset parity learning task as a testbed (see A.6 in the appendix). Note that the k subset parity learning task is notoriously hard to learn theoretically (see [2] for further details on this). In this setting, we clearly observe when using a small $\epsilon$:
>   1) Multiple Slingshot cycles clearly past the point of 100% training accuracy (figure 7).
>   2) Validation accuracy improves in a step like fashion with each Slingshot (figure 7).
>
> For larger values of epsilon, we observe neither 1)  nor 2) (figures 8 and 9 in the appendix)
>
> We believe this set of results is especially interesting to the machine learning community.
>
> **Questions**
>
> **Regarding Figure 3(d) (main paper):** We thank the reviewer for identifying and raising this issue. There is in fact a curvature spike where you expect it to be. We have spotted an alignment bug in Figure 3(d) in the main text of the paper. As a consequence, we have rerun the experiment and regenerated the figure (this issue has also been spotted by **Reviewer 8hZg**).
>
> **Regarding Figure 5 (main paper):** This plot shows test accuracy. All models were trained to 100% training accuracy (perfect fitting).
>
> **Dataset size** - As illustrated in figure 6 (main paper), Slingshots can happen in large datasets as well (we refer to the full CIFAR 10 dataset as “large” although that is debatable). However, we suspect clear generalization benefits might get somewhat diminished in large data settings (see our experiments on the full CIFAR10 dataset).
>
>
> ### References
>   - [1] Power et al - “Grokking: Generalization Beyond Overfitting on Small Algorithmic Datasets”
>   - [2] Barak et al - “Hidden Progress in Deep Learning: SGD Learns Parities Near the Computational Limit”.
>
> We kindly ask the reviewer to consider increasing their score if we have adequately addressed their concerns.

---

### Official Review · Reviewer_BLwe · 2022-07-18

**Rating:** 5
**Confidence:** 2
**Soundness:** 3 good
**Presentation:** 3 good
**Contribution:** 2 fair

**Summary:**

The paper presents an empirical study on an optimization anomaly, which is referred to as the Slingshot Mechanism/Effect by the authors, when adaptive optimizers (e.g., Adams) are used for solving supervised learning problems. Specifically, it was found that, when adaptive optimizers were used, cyclic phase transitions between stable and unstable training occurred and were correlated with how the norm of the last-layer weights changed. The authors also reported findings on the correlations between the onset of the Slingshot Mechanism and the glokking phenomenon.

**Questions:**

**Is the reported phenomenon unique to "deep models"?**

I did not see any numerical experiments on "shallow models." From the appendix, I only see experiments with a transformer (12 layers and 10 million parameters), a CNN with VGG-like architecture (which I assume to be a deep model), a deep MLP with 6 layers, and a deep linear model with 6 layers. Without experiments with simple/shallow models, it is premature to draw the conclusion that Slingshot is a general phenomenon as the authors claimed in Lines 154-162.

As a matter of fact, given that the authors tried to illustrate the phenomenon using simple quadratic functions (Line 174), could the authors conduct a set of numerical experiments using quadratic functions to see whether the phenomenon still persists?

**Some simple numerical diagnoses could have been done.**

As evident from Line 178, one issue with the adaptive optimization schemes considered is that the effective step size $\mu / (|g| + \epsilon)$ can be much larger than the unnormalized step size $\mu$ when the magnitude of the gradient $g$ is small and $\epsilon$ is small. (Here, a small $\epsilon$ is assumed; otherwise, the update rule will not adapt much to the magnitude of $g$.) This implies that, when the magnitude of $g$ is small, i.e., when the current iterate is near a stationary point, the next iterate of the gradient-based update can be very far from the stationary point, potentially leading to a sudden increase in the loss.

For gradient-based updates, when the objective function is smooth, it is known that a sufficient condition for decrease is to choose the step size to be less than $2/L$, where $L$ is the Lipschitz constant of the gradient (and is related to the Hessian). It would be useful to plot in Figure 3 the effective step size, given by $\mu / (|g| + \epsilon)$, along with the sharpness to see if the step size is still small enough for the sufficient condition for decrease to hold. If the violation of the condition coincides with the increase in the loss, then this would provide a simple explanation of the Slingshot Effect.

**Comments on presentation and typos:**

* Are the Slingshot Mechanism and the Slingshot Effect the same thing? Both terms are used throughout the paper.
* Figure 1 and its first citation are too far apart: Figure 1 appears on the first page (even before the main text), but it citation does not appear until Line 158.
* A typo in Line 190: "...evidence of **Elingshot** Effects."
* Line 230: What is $3.10e^{-04}4$? Also, it uncommon to write $10^{-04}$ instead of $10^{-4}$. This has occurred several times throughout the paper.

**Limitations:**

I do not have any comments on this.

**Strengths And Weaknesses:**

Strengths:

* The reported findings hint that extra care may be required when adaptive optimizers are used. The results may also promote further theoretical studies on the limitations of using adaptive optimizers in supervised learning.

Weaknesses:

* Additional numerical experiments could have been conducted to help future theorists better understand the source of issues. In particular, some rather simple diagnoses could have been done but were missing from the paper.
* There is much room for improvement in presentation of the paper. See the specific comments in the Questions section.

---

> ### Author Response · Authors · 2022-08-02
> **Response to Reviewer BLwe**
>
> We thank the reviewer for their review and comments. Please see our response bellow:
>
> **Is the reported phenomenon unique to "deep models"?**
>   - From our extensive empirical evaluations the phenomena is unique to models with more than 1 layer. The reason we have mainly focused on relatively deep models is that it seems depth is conducive to the effect, and it is more easily observable (more Slingshot cycles at earlier stages of training). However, we can confirm that the Slingshot effect is absolutely observable in 2 layer networks (but not 1).
> To address the reviewers concern, we have added an experimental section to the appendix (see appendix A.4, and specifically figure 5) illustrating the effect of depth on an MLP.
>
> **Regarding the L2 loss -**
>   - We do not observe the Slingshot effect when using the L2 loss. An intuitive explanation to this is that training a network with cross entropy in an unregularized setting will ultimately push the weights towards infinity, while for the L2 loss some set of finite weights achieve the optimum. Note that we only examine L2 loss as a means to undetstand how epsilon and local curvature interact during the optimization process. As stated in the paper (lines 171-172), in toy settings the epsilon parameter provably determines whether an adaptive optimizer will stably converge to the optimum. Hence we hypothesize that a small epsilon will cause a Slingshot effect in high curvature areas in parameter space (lines 186-189). This however is not a satisfactory causal explanation of the effect for the following reasons:
>       1) We do not see the Slingshot effect in 1 layer networks, hence depth seems to be a necessary condition.
>       2) We do not see the Slingshot behavior with the L2 loss — possibly because local curvature does not sufficiently increase.
>       3) This does not explain the cyclic nature of divergence - recovery as described and demonstrated in the paper, and
>       4) This does not explain the apparent benefit to generalization.
>
> **Some simple numerical diagnoses could have been done.**
> - We appreciate the suggestion and will add an experimental section discussing this diagnostic in future versions. However we respectfully disagree with the statement that it would “provide a simple explanation for the Slingshot Effect”. For one, from the spiking loss the local curvature in the direction of the update trivially cannot accommodate the effective learning rate, hence estimating the Lipschitz constant would not provide much insight. Moreover, this does not explain the training dynamics which give rise to such a sudden curvature spike, and consequently Slingshots as we see in our experiments for the reasons given above (see our response regarding the L2 loss) . As a matter of fact, a simple way of removing all Slingshot effects would be to increase epsilon. An unfortunate side effect of this is a hit in performance.
> Additionally, as the reviewer suggested, some notable theoretical papers have proven guarantees of convergence for Adam under some restrictions. Notably, in [1] and [2], a learning rate of $O(\frac{1}{\sqrt(t)})$ ($t$ is number of iterations) is needed for convergence, where the proofs rely on a non increasing effective learning rate. This regime of convergence is clearly suboptimal and is at odds with our observations: The cyclic divergence and convergence behavior we have uncovered always leads to improved performance in our experiments, and present a unique convergence characteristic that has yet to be discussed in the literature as far as we can tell. See line 196-208 of section 3 and Figure 4 in the main paper.
>
> **Comments on presentation and typos:**
> - We further thank the reviewer for their comments on presentation and typographical errors. We have fixed the typographical errors brought to our attention by the reviewer. We will address any remaining errors in the final version of the paper.
>
> ### References:
>   - [1] Reddi et al - “On the convergence of Adam and beyond”
>   - [2] Defossez et al - “A Simple Convergence Proof of Adam and Adagrad”
>
> We hope we have adequately addressed the reviewers concerns, in which case we kindly ask that they consider increasing their score.

---

> > ### Comment · Reviewer_BLwe · 2022-08-05
> > **More questions on the numerical diagnoses**
> >
> > I would like to thank the authors for adding additional numerical experiments that may help future theorists investigate the phenomenon. I have a few additional questions regarding the authors' response on the numerical diagnoses that I requested in my previous comment:
> >
> > 1. The authors claimed that "from the spiking loss the local curvature in the direction of the update trivially cannot accommodate the effective learning rate." Could you clarify what "accommodate" refers to in this context? Also, how does the spiking loss directly relate to the local curvature?
> > 2. Could the authors elaborate on why "estimating the Lipschitz constant would not provide much insight"? As mentioned in my previous comment, the Lipschitz constant of the gradient would provide a sufficient condition for the learning rate/step size to induce a decrease in the loss.
> > 3. The authors claimed that examining effective step size "does not explain the training dynamics which give rise to such a sudden curvature spike." Could you elaborate on why? From my perspective, if the effective step size is too large (compated to $2/L$, where $L$ is the local Lipschitz constant of the gradient), then the next iterate can be very far away from the current iterate and may land somewhere with high loss by chance. That was why I believed comparing the effective step size with the local Lipschitz constant of the gradient would be useful.

---

> > > ### Author Response · Authors · 2022-08-07
> > > **Response to More questions on the numerical diagnoses**
> > >
> > > *We thank **reviewer BLwe** for their active engagement and thoughtful questions.*
> > >
> > > We first apologize for confusing points made by the paper and our previous responses. We want to clarify that we agree with the reviewer that large Lipschitz L would indeed lead to the loss spiking. However, we meant to point out that this Lipschitz based analysis does not explain the full Slingshot Effect, namely the cyclic loss/weight norm behaviors and their implication on generalization. See detailed responses below.
> > >
> > > 1. As the reviewer pointed out, under gradient based optimization and some regularity conditions, a sufficient condition for the loss to decrease is to set the effective learning rate to be lower than 2/L. From the sufficiency of the 2/L condition, and the spiking loss at the phase transitions of the Slingshots, we can conclude that the effective step size at the phase transitions must indeed be larger than 2/L. Since L represents the local Lipschitz of the gradient in a local region, it is given by the maximum eigenvalue of the Hessian in the region, or maximum curvature. By “cannot accommodate” we simply mean that the effective learning rate is in fact too high.
> > > 2. By “estimating the Lipschitz constant would not provide much insight” we mean that it is given that the 2/L condition on the learning rate is violated (as you suggested), otherwise we wouldn’t see a spiking loss. That being said, we ran additional experiments to measure the effective step size, as well as an estimate of the local Lipschitz of the gradients. This is given in figure 10 in the appendix. Observe that the effective stepsize is indeed larger than the maximum allowed according to our estimations (see details in appendix A.6.1). Moreover, after a few cycles of slingshots, it appears the Lipschitz constant drops dramatically, indicating convergence to a low curvature minima (this is also hypothesized in the paper - line 186 - 187). We thank the reviewer for suggesting this experiment.
> > > 3. We agree with the reviewer’s perspective that the high Lipschitz constant might be a direct cause of the divergence behavior. However, the spiking loss is only half of the Slingshot story. A staple of the Slingshot Effect is the cyclic transitions between divergence and convergence of the loss, and improved generalization. Currently no adequate explanation exists to why the optimization trajectory repeatedly enters these regions of high local Lipschitz.  More importantly, our results suggest that it is not optimal to avoid Slingshots in terms of test performance (see figures 5,6 in the main paper, and 7 in the appendix). Note that this is clearly at odds with the typical analyzed setting of stable, monotone convergence ([1],[2]) and at a minimum points to a rethinking of the theory.
> > >
> > >
> > > ### References:
> > > * [1] Reddi et al - “On the convergence of Adam and beyond”
> > > * [2] Defossez et al - “A Simple Convergence Proof of Adam and Adagrad

---

### Author Response · Authors · 2022-08-02
**General Response and Manuscript Updates**

We thank all reviewers for carefully reading our manuscript and providing positive and thoughtful feedback.
We have updated our paper and appendix in order to address the concerns that were raised. In addition, we will be providing code for the community to reproduce and extend our work.

**Issues with Plots:**
  - Figure 2(c): **Reviewer 8hZg** kindly pointed out an issue with Figure 2(c). This plot has been updated to fix an issue with a mislabeled plot in our initial draft. Note that a correct version of validation loss was available in Figure 22 in the appendix submitted for review.
  - Figure 3: **Reviewer KF2y** and **Reviewer 8hZg** pointed out missing curvature spikes in Figure 3(d) specifically. We have spotted an alignment bug in Figure 3(d). As a consequence, we have rerun the experiment and regenerated the figures for all datasets in these experiments.

**Shallow(er) models:**
  - **Reviewer Blwe** pointed out the lack of Slingshot demonstration with shallow(er) models. We have added additional experiments with 1, 2 and 3-layer networks in the appendix A.4 and Figure 5 in the appendix. We find evidence of Slingshot Effects with 2 and 3-layer models while there is no evidence of Slingshot with the 1-layer model.

**Additional experiments:**
  - To provide further evidence of a potential correlation between the Slingshots and generalization and to demonstrate its generality, we have added new additional experiments on learning k sparse parities task. These experiments are different from the tasks used in the originally submitted paper, borrowing the setting from Barak et al. [1] (potentially interesting to theoreticians, for more details refer to [1]). We find that this testbed exemplifies the Slingshot effect as described in our paper. The results are consistent with our observations already in the paper, namely multiple Slingshot cycles with improved validation accuracy in a staircase manner. We thank **Reviewer KF2y** for motivating us to show how Slingshots might be conducive for generalization in additional settings.

[1] Barak et al - “Hidden Progress in Deep Learning: SGD Learns Parities Near the Computational Limit”.

---

### Meta-Review · Area_Chair_mztE · 2022-08-27

**Recommendation:** Reject
**Confidence:** Less certain

**Metareview:**

The paper examines a widely known phenomenon when training neural networks with adaptive optimizers, where the training loss cyclically alternates in later stages. Some evidence is given that, in the absence of explicit regularization, this is associated with improved generalization. The more concrete contribution of the paper is to show a strong link between the aforementioned cyclicity and sudden cyclic growth in the weight of the last layer of the network. The main concern is that beyond this empirical observation a specific mechanism behind the phenomenon is not identified, nor a clear connection is made with the apparent benefit to generalization. While the observations have merit, it is hard to determine whether cause-effect relationships exist between them, making the paper feel as a work-in-progress and only weakly significant to the community (judging by the reviewers being underwhelmed). If the authors are convinced of their message that "slingshots" cause "grokking" (in contrast to, e.g., being a by-product while the real mechanism lies elsewhere), then they are advised to show exactly that in the further elaboration of their work.

**Award:**

No

---

### Decision · Program_Chairs · 2022-09-14

Reject